# Structural basis of homo- and heterotrimerization of collagen I

Urvashi Sharma[1,*], Loïc Carrique[1,*], Sandrine Vadon-Le Goff[2,*], Natacha Mariano[2], Rainier-Numa Georges[2], Frederic Delolme[2,3], Peppi Koivunen[4], Johanna Myllyharju[4], Catherine Moali[2], Nushin Aghajari[1,**] & David J.S. Hulmes[2,**]

Fibrillar collagen molecules are synthesized as precursors, procollagens, with large propeptide extensions. While a homotrimeric form (three α1 chains) has been reported in embryonic tissues as well as in diseases (cancer, fibrosis, genetic disorders), collagen type I usually occurs as a heterotrimer (two α1 chains and one α2 chain). Inside the cell, the role of the C-terminal propeptides is to gather together the correct combination of three α chains during molecular assembly, but how this occurs for different forms of the same collagen type is so far unknown. Here, by structural and mutagenic analysis, we identify key amino acid residues in the α1 and α2 C-propeptides that determine homo- and heterotrimerization. A naturally occurring mutation in one of these alters the homo/heterotrimer balance. These results show how the C-propeptide of the α2 chain has specifically evolved to permit the appearance of heterotrimeric collagen I, the major extracellular building block among the metazoa.

[1] Molecular Microbiology and Structural Biochemistry Unit, UMR 5086 CNRS - University of Lyon 1, 7 passage du Vercors, F-69367 Lyon, France. [2] Tissue Biology and Therapeutic Engineering Unit, UMR 5305 CNRS - University of Lyon 1, 7 passage du Vercors, F-69367 Lyon, France. [3] SFR Biosciences - Protein Science Facility, University of Lyon 1, Ecole Normale Supérieure de Lyon, INSERM US8, CNRS UMS 3444, 50 Avenue Tony Garnier, F-69366 Lyon, France. [4] Oulu Center for Cell-Matrix Research, Biocenter Oulu and Faculty of Biochemistry and Molecular Medicine, University of Oulu, P.O. Box 5000, FI-90014 Oulu, Finland. * These authors contributed equally to this work. ** These authors jointly supervised this work. Correspondence and requests for materials should be addressed to N.A. (email: nushin.aghajari@ibcp.fr) or to D.J.S.H. (email: david.hulmes@ibcp.fr).

The extracellular matrix (ECM) comprises over 1,000 proteins that together provide both mechanical support and signalling functions in almost all multicellular organisms[1]. One of the main families of ECM proteins are the collagens (currently 28 different types in humans) among which the fibrillar collagens (types I, II, III, V and XI) constitute the classical periodically banded fibrils seen by electron microscopy[2]. Fibrillar collagen molecules are synthesized in precursor form, procollagens, consisting of a rod-like central triple-helical region (~300 kDa) with globular propeptide extensions at both the N- (~50 kDa) and C- (~90 kDa) termini (Fig. 1). Each procollagen molecule is a trimer, where, depending on the genetic type, the three polypeptide chains are either identical (homotrimers, as in collagens II and III) or at least one of the chains is distinct (heterotrimers, as in collagens I, V and XI). Collagen I is the most widely occurring type, being the major extracellular constituent of skin, bone, tendons, cornea, lung, heart and blood vessels. Though this collagen mostly occurs as a heterotrimer with two α1(I) chains and one α2(I) chain, small amounts of a homotrimeric form (three α1(I) chains) have been reported in adult skin[3] as well as greater levels in embryonic tissues[4]. Increased quantities of collagen I homotrimer are also associated with cancer[5–9], fibrosis[10–12], periodontal disease[13], osteoarthritis[14], osteoporosis[15], osteogenesis imperfecta (OI)[16–18] and Ehlers–Danlos syndrome[19]. The homotrimeric form differs in properties compared to the heterotrimeric form, notably by its impaired ability to form fibrils[20–22] and by its increased resistance to proteinases[23,24]. Like collagen I, collagen V also exists in homotrimeric and heterotrimeric forms, which appear to have different functions *in vivo*[25]. However, the molecular mechanisms determining homo- or heterotrimerization of these fibrillar collagens remain unknown.

Inside the cell, within the endoplasmic reticulum, the C-propeptides (also called COLFI domains) play key roles in determining correct chain association during trimerization, notably in cells producing different collagen types[26]. For the fibrillar procollagens, once the three chains have folded to form the C-propeptide trimer, folding of the triple-helical region continues in a zipper-like manner towards the N-terminus[27]. While in general the amino acid sequences of the C-propeptides are highly conserved, Lees et al.[26] identified a highly variable so-called chain recognition sequence (CRS) which appeared to be required for specific chain recognition during trimer assembly. The CRS is discontinuous consisting of a long stretch (12 residues; CRS long) and a short stretch (3 residues; CRS short) separated by a relatively conserved sequence of 8 residues. We recently revealed the structural basis of chain recognition by determining the first three-dimensional (3D) structure of a C-propeptide trimer, that of human procollagen III (CPIII), an obligate homotrimer[28]. Within this structure, the two parts of the CRS appeared on opposite sides of each interaction interface between adjacent chains, thus explaining the specificity of chain association.

The 3D structure of CPIII raised the question of how chain association is controlled in the C-propeptides of procollagen I (CPI), which can occur in both heterotrimeric and homotrimeric form. Here we present the crystal structure of the homotrimeric form of CPI (homo-CPI) determined to 2.2 Å resolution which shows striking differences in the inter-chain interface compared to CPIII. We also present data on the corresponding heterotrimer (hetero-CPI) that reveal how a naturally occurring missense mutation in the α1(I) C-propeptide alters the homotrimer/ heterotrimer balance and how the sequence of the α2(I) C-propeptide is perfectly adapted to the formation of the [α1(I)]₂α2(I) heterotrimer.

## Results

**CPI homotrimer.** Homo-CPI was expressed in mammalian cells and purified as for CPIII (ref. 29). The purified protein showed a single band by SDS-PAGE with masses of ~85 kDa in non-reducing conditions and 30 kDa in reducing conditions corresponding to the disulfide-bonded trimer (Supplementary Fig. 1). By mass spectrometry, the observed mass was $85{,}824 \pm 108$ Da compared to an expected mass of 85,932 Da. Crystals grown belonged to space group $P2_1$ and diffracted X-rays to 2.2 Å. Structure determination was by molecular replacement with refinement to the same resolution (data collection and refinement statistics in Table 1).

The overall structure of the homo-CPI trimer strongly resembles that of CPIII having the shape of a flower, with a stalk, a base and three petals (Fig. 2). There are two trimers in the asymmetric unit, the structures of which are very similar (root mean square deviation (r.m.s.d.) = 0.67 Å), with only slight movements in the stalk and petal regions. The stalk consists of an α-helical three-stranded coiled-coil formed by residues 8–29. Residues 30–76 form the base region, which is particularly

## Table 1 | Data collection and refinement statistics.

| | |
|---|---|
| PDB-ID | 5K31 |
| *Data collection* | |
| Space group | $P2_1$ |
| Cell dimensions | |
| $a, b, c$ (Å) | 74.82, 149.63, 105.95 |
| $\beta$ (°) | 101.70 |
| Resolution range (Å) | 47.07–2.20 (2.24–2.20) |
| Total no. of reflections | 369,072 (17,277) |
| Unique reflections | 115,206 (5,630) |
| $R_{merge}$ (%) | 7.8 (38.9) |
| $I/\sigma(I)$ | 7.7 (2.3) |
| Completeness (%) | 99.8 (99.6) |
| Redundancy | 3.4 (3.1) |
| No. mol. /asymm. unit | 6 (2 trimers) |
| | |
| *Refinement* | |
| $R_{work}/R_{free}$ (%) | 19.5 / 23.8 |
| No. atoms | |
| Water | 420 |
| Protein | 11,083 |
| Ligand | 14 |
| Calcium | 6 |
| Chloride | 3 |
| Average $B$-factor (Å²) | 41.47 |
| Protein | 42.03 |
| Ligand/ion | 43.60 |
| Water | 26.38 |
| r.m.s.d. | |
| Bond lengths (Å) | 0.019 |
| Angles (°) | 1.942 |

r.m.s.d, root mean square deviation.
Values in parentheses are for the highest resolution shell.

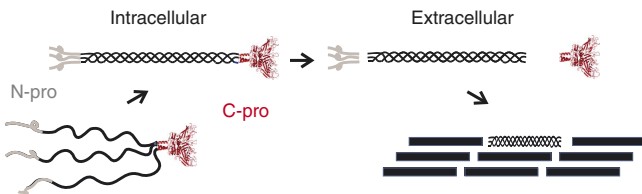

**Figure 1 | Roles of the procollagen C-propeptides.** Trimerization of the procollagen C-propeptides initiates intracellular assembly of the procollagen molecule while extracellular proteolytic cleavage of the N- and C-propeptides controls collagen fibril formation.

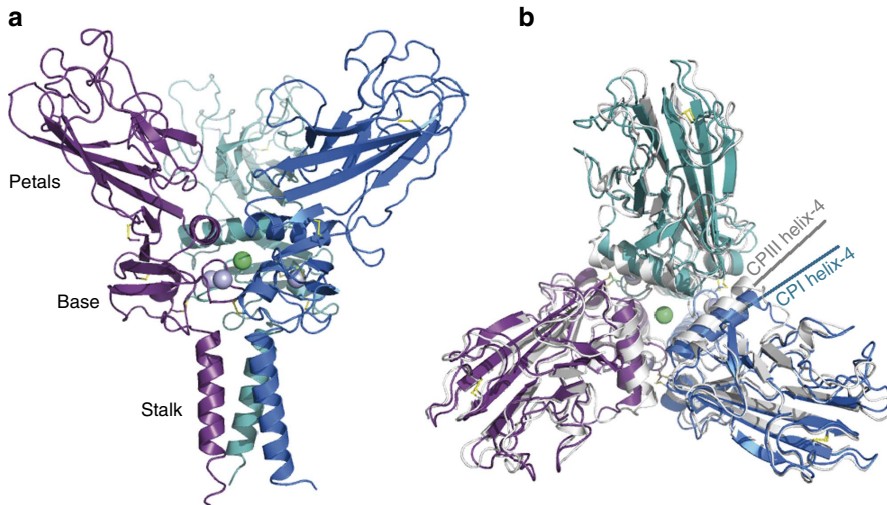

**Figure 2 | 3D structure of homo-CPI.** (**a**) View from the side showing the stalk, base and petal regions. Each chain is represented in a different colour, with bound $Ca^{2+}$ ions as light blue spheres and the $Cl^-$ ion in green. Disulfide bonds are in yellow. (**b**) View from the top showing a structural alignment of the homo-CPI structure (in colour) on the 3.3 Å structure of CPIII (PDB code 4AK3; in grey). While overall the two structures are well aligned, there is a shift in orientation of helix-4 (highlighted for one chain from each structure).

highly conserved among the different fibrillar procollagens (Supplementary Fig. 2A). This region includes three tightly bound $Ca^{2+}$ ions (one per chain; same coordination as in CPIII) and one $Cl^-$ ion in the centre coordinated by Gln62 and Arg39 from all three chains, as well as one intra-chain (Cys41–Cys73) and one inter-chain (Cys47–Cys64) disulfide bond. The remainder of the molecule constitutes the petal region (residues 77–246) which is stabilized by two intra-chain disulfide bonds (Cys81–Cys244, Cys152–Cys197) and contains the long and short stretches of the CRS.

While the overall conformation of homo-CPI is very similar to that of CPIII (Fig. 2), an r.m.s.d. value of 1.35 Å reveals some structural rearrangements. The main difference is in the orientation of helix-4, a relatively long α-helix in each chain whose ends are involved in the inter-chain interactions thus forming a central triangle in the trimer (Fig. 2b). While the rest of CPI aligns well with CPIII (except at the extremities of the petal regions) there is a clear difference in the orientation of helix-4. This is also seen when individual chains of CPI and CPIII are aligned (r.m.s.d. = 0.89 Å; Supplementary Fig. 2B). In addition, there are striking differences in the interaction interfaces between neighbouring chains (Fig. 3a,b; Supplementary Movies 1 and 2; Supplementary Table 1). While the inter-chain interface in CPIII is characterized by a relatively large number of specific interactions (salt bridges and hydrogen bonds) involving residues in the CRS[28], there are surprisingly few of these interactions in homo-CPI. The only example involving the CRS is the salt bridge between Asp129 (CRS long) and the highly conserved Arg42 situated in helix-2 in the base region (Fig. 3b, also Supplementary Fig. 3). Arg42 also forms a salt bridge with the conserved Asp67 in the base region, which moreover is involved in $Ca^{2+}$ coordination. Comparison of the salt bridges involving Arg42 in CPIII (with Asp127) and in homo-CPI (with Asp129) shows a large swing ($\sim$5 Å) in the side-chain of Arg42 (Fig. 3a,b). Furthermore, again unlike in CPIII, in homo-CPI, apart from relatively non-specific interactions involving conserved hydrophobic and polar residues, there are no specific interactions involving the short stretch of the CRS. Indeed, there are very few inter-chain interactions involving residues from anywhere in the petal region in homo-CPI, unlike in CPIII which contains several such interactions involving both CRS and non-CRS residues (Supplementary Table 1). Instead, homo-CPI

seems to be stabilized by interactions mainly involving residues in the base region.

Further differences between homo-CPI and CPIII relate to the asymmetry in the structure previously described for CPIII (ref. 28). This was first revealed by structural alignment showing a difference in conformation in one of the three chains (despite sequence identity) in the short stretch of the CRS. Asymmetry in CPIII is also evident by detailed analysis of the interactions at the three inter-chain interfaces (Supplementary Table 1). This shows a relatively large number of interactions at the A:B chain interface (compared to B:C and A:C) as well as more subtle differences such as a switch in interaction partner for Asp127 from Arg42 (in the base region) at the A:B and B:C interfaces to Arg142 (in CRS short) at the A:C interface. In contrast, structural alignment of the three chains in homo-CPI reveals little sign of asymmetry (r.m.s.d. = 0.34 Å), albeit that minor differences are detectable in the inter-chain interactions.

To further investigate the contributions of individual residues to stabilizing the inter-chain interface, we used a site-directed mutagenesis approach. We began with residue Asp129 which appears to play a crucial role in inter-chain interactions. Surprisingly, when conditioned medium was analysed by SDS-PAGE and western blotting, trimerization of His-tagged homo-CPI was unaffected by the mutations Asp129Asn or Asp129Ala (Fig. 4ai). Closer inspection of the structure showed that the side-chain of Asp126 is close to that of Asp129 (Fig. 3b) and therefore might substitute for Asp129 when the latter is absent, thus maintaining the salt bridge with Arg42. As for Asp129, the single-mutations Asp126Asn and Asp126Ala had little effect on trimerization. In contrast, the double-mutant Asp126Ala/Asp129Ala was almost completely unable to trimerize (Fig. 4ai).

**CPI heterotrimer**. We also expressed hetero-CPI, which consists of two α1 chains and one α2 chain, by co-transfection with a 2:1 ratio of plasmids encoding the α1 and α2 C-propeptides. An N-terminal His-tag on the α2 chain permitted separation from untagged homo-CPI that was also synthesized and secreted into the culture medium. We also produced a form with a cleavage site for tobacco etch virus (TEV) protease just after the His-tag in the α2 chain thus allowing removal of the His-tag after separation from homo-CPI. Following gel filtration, we obtained pure

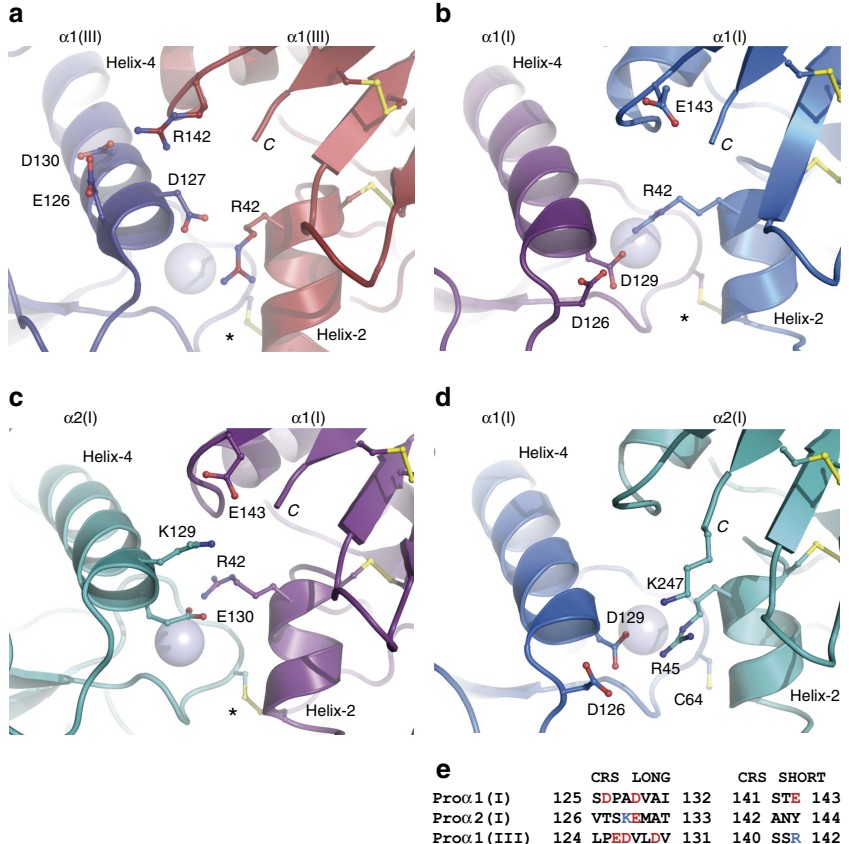

**Figure 3 | Interaction interfaces in different C-propeptide trimers.** (**a**) Charged residues at the A:B chain interface in the 1.7 Å structure of CPIII (PDB code 4AE2). Each chain is shown in a different colour, as in Bourhis et al.[28]. (**b**) Charged residues at the inter-chain interface in homo-CPI. Chains are coloured as in Fig. 2. Note the large conformational change in the side-chain of residue Arg42 between CPIII and homo-CPI. (**c,d**) Models of the two inter-chain interfaces in hetero-CPI involving the α2(I) chain. Chains are coloured as in Fig. 2, with the chain in deep teal (greenish blue) replaced by the α2(I) chain. C denotes the C-terminus of each chain and * indicates the position of the inter-chain disulfide bond, which is absent in **d** leaving the free cysteine Cys64. (**e**) Sequence alignment in the CRS region for CPI (α1 and α2 chains) and CPIII. See also Supplementary Movies 1–4.

hetero-CPI as shown by SDS-PAGE (Supplementary Fig. 1) with a single band at ∼85 kDa under non-reducing conditions, corresponding to the trimer, which appeared as a double band after reduction at around 30 kDa, showing the α1 and α2 chains in an approximate 2:1 ratio. By mass spectrometry, the observed masses of hetero-CPI were 84,693 ± 106 Da (with His-Tag) and 83,641 ± 105 Da (without His-tag), in good agreement with the expected masses of 84,617 Da and 83,584 Da, respectively.

Unfortunately, despite numerous crystallization trials with both forms of hetero-CPI, we were unable to obtain any crystals suitable for crystallography. By circular dichroism, the far UV spectrum of hetero-CPI was very similar to that of homo-CPI, both being similar to that of CPIII (Supplementary Fig. 4A), thus indicating no major differences in secondary structure between all three C-propeptide trimers. To determine the low resolution structure of hetero-CPI, we analysed the protein in solution by small angle X-ray scattering (SAXS). This resulted in a linear Guinier plot (Supplementary Fig. 5A) corresponding to a radius of gyration of 33.4 Å and an observed mass (calculated from the zero angle intercept) of 86 kDa. Also, the maximum dimension in the distance distribution function (p(r)) was ∼110 Å and the Kratky plot showed the typical profile of a well-folded multi-domain protein[30,31] (Supplementary Fig. 5B,C). These parameters are in excellent agreement with the data previously obtained for CPIII (ref. 32) and compare well with the radius of gyration (31.4 Å) and maximum dimension (100 Å) calculated from the crystal structure of homo-CPI. Finally, the scattering pattern of

hetero-CPI was strikingly similar to that calculated from the crystal structure of homo-CPI ($\chi^2 = 4.97$; Supplementary Fig. 5D). Taken together, these data showed that the low resolution structure of hetero-CPI was similar to that of homo-CPI and therefore that the recombinant system could be used for studying the effects of mutations on heterotrimer assembly.

As mentioned earlier, from the crystal structure, the short stretch of the CRS in homo-CPI is not involved in inter-chain salt bridge interactions. The sequence of CRS short includes the charged acidic residue Glu143, but this lacks a possible binding partner in the long stretch of the CRS in an adjacent α1(I) chain (Fig. 3b). However, examination of the long stretch of the CRS in the α2(I) chain reveals the presence of the basic charged residue Lys129 at the position corresponding to Ala128 in the α1(I) chain (Fig. 3e). Modelling of the α2(I) chain in the heterotrimer based on the structure of homo-CPI suggests that α2(I)-Lys129 is ideally positioned to form a salt bridge with Glu143 in the short stretch of the α1(I) CRS (Fig. 3c; Supplementary Movie 3). Such an interaction between the α1(I) and α2(I) chains might therefore contribute to the stabilization of hetero-CPI. In this regard, there is a naturally occurring missense mutation in the *COL1A1* gene, corresponding to Glu143Lys in the α1(I) C-propeptide, which is associated with the brittle bone disorder OI type IV (ref. 33). From the structure-based modelling, we predicted that this mutation would destabilize the heterotrimer and thus favour the formation of homo-CPI. To test this hypothesis, we first expressed His-tagged homo-CPI containing the Glu143Lys

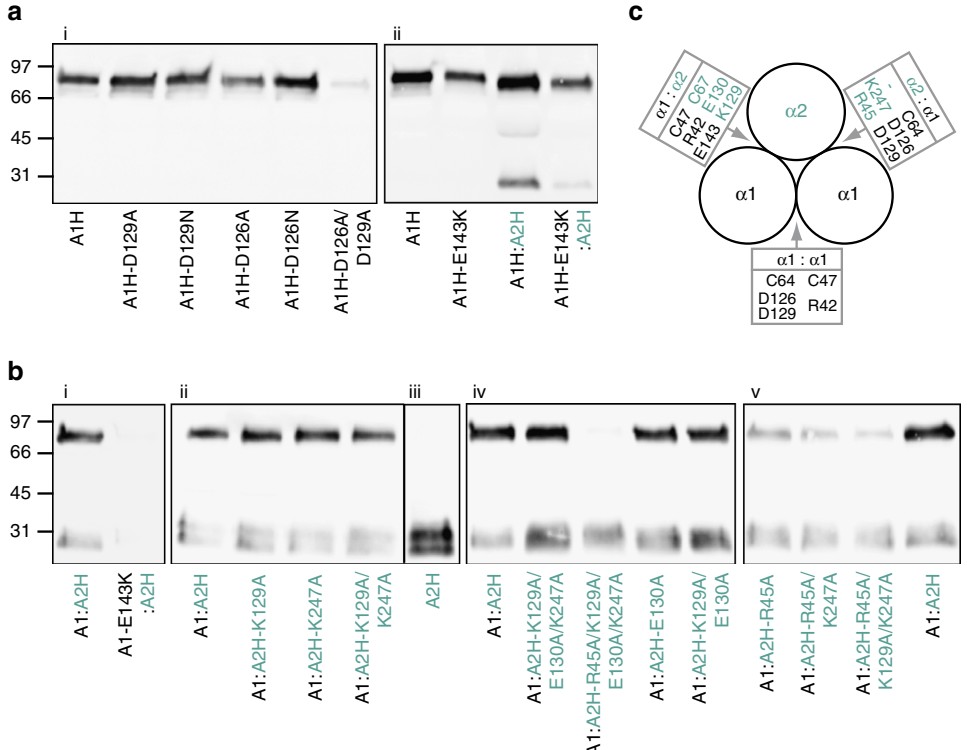

**Figure 4 | Effects of site-directed mutations on the trimerization of homo-CPI and hetero-CPI.** Proteins secreted into the culture medium after site-directed mutagenesis of the residues indicated were detected by western blotting using the N-terminal His-tag present on either or both the α1(I) and α2(I) chains, as indicated by H (for example, A1H means His-tagged α1 chain). The α2(I) chain is indicated by the green/blue colour. Gels were run in non-reducing conditions, where trimers migrate at 85 kDa and monomers at 30 kDa. (**a**) Controls and mutants using the His-tagged α1(I) chain. (**b**) Controls and mutants using the non-tagged α1(I) chain and the His-tagged α2(I) chain. (**c**) Schematic view down the axis of a C-propeptide heterotrimer showing interacting residues at the homo- and hetero-chain interfaces. Same colour coding throughout. Data shown are representative of triplicate (**a**) or duplicate (**b**) biological replicates.

mutation in the α1(I) C-propeptide. As expected, compared to the wild-type protein, this mutation had little effect on expression of the His-tagged homo-CPI (Fig. 4aii). Similar results were obtained when both the α1(I) and α2(I) chains were His-tagged, where the 85 kDa band includes both homotrimers and heterotrimers (Fig. 4aii; Supplementary Fig. 6). To determine the effect solely on heterotrimer formation, we then co-expressed the non-tagged α1(I) chain containing the Glu143Lys mutation with the wild-type His-tagged α2(I) chain. When analysed by western blotting, compared to the wild-type heterotrimer, the Glu143Lys mutation in the α1(I) chain led to a strong reduction in incorporation of His-tagged α2(I) chain into heterotrimers (Fig. 4bi), as predicted by structure-based modelling. Thus a missense mutation in the α1(I) chain has little effect on homotrimer formation but diminishes incorporation of the wild-type α2(I) chain into the heterotrimer.

It is well known that the α2(I) C-propeptide on its own is incapable of forming homotrimers[26], due to the absence of one of the cysteine residues (Cys50) involved in inter-chain disulfide bonding (Supplementary Fig. 2A). This was confirmed when the His-tagged wild-type α2(I) chain was expressed alone (Fig. 4biii). It should also be noted that small amounts of monomeric α2(I) C-propeptide were often found in the medium when co-expressed with the non-tagged α1(I) C-propeptide (Fig. 4b).

According to the model, in addition to the salt bridge between α1(I)-Glu143 and α2(I)-Lys129, there is a salt bridge between α1(I)-Arg42 and α2(I)-Glu130 (Fig. 3c), analogous to that between α1(I)-Asp129 and α1(I)-Arg42 in homo-CPI (Fig. 3b). In the hetero-CPI trimer, these interactions occur in one of the

two interfaces involving the α2(I) chain which is bound to two α1(I) chains (Fig. 4c). The other interface (Fig. 3d; Supplementary Movie 4) involves the C-terminal residue of the α2(I) chain. As previously noted for CPIII (ref. 28), and also seen here for homo-CPI (Fig. 3b), the C-terminal residue is found close to the inter-chain interaction interface. While this is a leucine residue for the other major fibrillar procollagen chains, in the α2(I) chain it is a lysine (Lys247). Modelling of the corresponding inter-chain interface in hetero-CPI (Fig. 3d) suggests a salt bridge between α2(I)-Lys247 and α1(I)-Asp126, in addition to one between α1(I)-Asp129 and α2(I)-Arg45, the latter residue being equivalent to α1(I)-Arg42.

To test the above hypotheses, we used site-directed mutagenesis to alter the following residues in the α2(I) chain to alanines: Arg45, Lys129, Glu130 and Lys247. His-tagged α2(I) chains containing the mutations were then co-expressed with untagged α1(I) chains to determine their effects on heterotrimerization (Fig. 4b). Concerning the individual mutations, Lys129Ala (Fig. 4bii), Lys247Ala (Fig. 4bii) and Glu130Ala (Fig. 4biv) had no effect on incorporation of the α2(I) into heterotrimers, as compared to the wild-type chain. The Arg45Ala mutation however (Fig. 4bv) resulted in a clear diminution in the amount of heterotrimer present in the medium, showing α2(I) Arg45 to be a key residue in heterotrimer formation. Concerning the double mutations, Lys129Ala/Lys247Ala (Fig. 4bii) and Lys129Ala/Glu130Ala (Fig. 4biv) had no effect, compared to wild-type, while Arg45Ala/Lys247Ala (Fig. 4bv) resulted in even less heterotrimer than Arg45Ala alone. This demonstrates the crucial role of the interface containing α2(I) Arg45 and α2(I)

Lys247, which lacks an inter-chain disulfide bond. We also made triple mutations, among which Lys129Ala/Glu130Ala/Lys247Ala (Fig. 4biv) had no effect on the amount of heterotrimer compared to wild-type. On the other hand, with Arg45Ala/Lys129Ala/Lys247Ala (Fig. 4bv) the amount of heterotrimer was even less than for Arg45Ala/Lys247Ala, such that there was a progressive reduction in heterotrimer with the single, double and triple mutations involving α2(I) Arg45. Finally, for the quadruple mutation Arg45Ala/Lys129Ala/Glu130Ala/Lys247Ala (Fig. 4biv), production of heterotrimer was completely abolished. These results demonstrate the crucial importance of the α2(I) chain residues Arg45, Lys129, Glu130 and Lys247 in heterotrimer formation. Thus the combination of structure-based modelling and site-directed mutagenesis shows that the α2(I) chain brings additional interactions to the interaction interfaces that may favour the assembly of heterotrimers versus homotrimers.

To look for differences in thermal stability, we collected circular dichroism data over the temperature range 25 to 85 °C. For CPIII, gradual changes were seen throughout most of the spectrum (Supplementary Fig. 4B) but particularly at 208 nm corresponding to one of the characteristic minima for α-helical structures[34]. On cooling down to 25 °C, however, the spectrum was hardly changed, showing this structural transition to be irreversible. When measured at a single wavelength of 208 nm (Supplementary Fig. 4C) both CPIII and hetero-CPI melted at ~62 °C while homo-CPI melted at ~69 °C.

## Discussion

The structure of homo-CPI presented here provides a structural explanation for why it is that different fibrillar collagens can be synthesized in the same cell without mixing up of α chains from different genetic types, such as for example collagens I and III. While overall the 3D structures of homo-CPI and CPIII are very similar, there are striking differences in the interaction interfaces between chains. In CPIII these are dominated by specific interactions mostly involving the long and short stretches of the CRS in the petal region, with relatively few interactions involving the base region. In contrast, in homo-CPI, residues from the CRS are hardly involved in the interface interactions, which occur almost entirely in the base region. Thus despite many residues being strongly conserved in both CPI and CPIII, residues that are specific to each contribute to avoiding the production of CPI:CPIII chimeric trimers. In homo-CPI, the only specific interaction involving the petal region is between Asp129 in the long stretch of the CRS with the conserved Arg42 in the base region. But even Asp129 is not indispensable for trimerization, since we show here that the nearby Asp126 is apparently able to take its place when Asp129 is mutated to Ala. In the crystal structure, Asp126, Asp129 and Arg42 are aligned in a row with Asp129 interacting with Arg42, thus preventing interaction with Asp126 (Supplementary Movie 2). In the absence of the Asp129 side-chain, a conformational re-arrangement may occur that allows Asp126 to bind to Arg42. Alternatively, this region of the molecule may occur in two different conformations, one of which permits direct binding of Asp126–Arg42, even in the presence of Asp129. If so, only the 'Asp129' conformation is represented in the crystals analysed here.

Analysis of the structure of homo-CPI gives insights into the interactions that might stabilize the heterotrimeric form of CPI. In particular, in the long stretch of the CRS, Ala128 in the α1(I) chain is replaced by Lys129 in the α2(I) chain, which appears to form a salt bridge with Glu143 in the short stretch of the CRS in a neighbouring α1(I) chain (Fig. 3c). This latter residue has been implicated in a missense heterozygous mutation (E143K or p.E1361K) associated with OI type IV. Reproducing this mutation

in the α1(I) chain using the hetero-CPI expression system markedly reduced the synthesis of heterotrimers, with little effect on homotrimer assembly, consistent with an attractive interaction between WT α2(I) Lys129 and α1(I) Glu143 being replaced by a repulsive interaction with the mutant α1(I) Lys143. This result is analogous to a number of other naturally occurring C-propeptide mutations (both missense and frameshift), this time in the procollagen α2(I) chain, that lead to different forms of OI and are associated with increased amounts of homotrimer[16–18,35]. To our knowledge, however, the Glu143Lys mutation in the C-propeptide of the α1(I) chain is the first example of a missense mutation in one chain that interferes with incorporation of a normal but genetically distinct chain into the trimer. In general, missense mutations in the C-propeptide region of the proα1(I) chain result in more severe forms of OI than those in the proα2(I) chain[35]. This is largely because 75% of procollagen molecules will have at least one mutant proα1(I) chain in heterozygotes, compared to only 50% for missense mutations in the proα2(I) chain. The situation is exacerbated in homozygotes however, as in a patient with a 4 base pair frameshift mutation in the proα2(I) chain that results in replacement of the C-terminal 33 amino acids leading to the complete absence of heterotrimers and severe OI type III (ref. 17). Since this mutation occurs in the last exon of the gene it escapes nonsense mediated mRNA decay; the mutant proα2(I) chain is therefore translated but cannot be incorporated into trimers. It is likely that the stress associated with the intracellular accumulation of these mutant chains is responsible for the OI phenotype[35]. Whether the accumulation of normal proα2(I) chains in the case of the proα1(I) p.E1361K (E143K) mutant can also contribute to an OI phenotype, especially in heterozygotes, is unknown.

Additional considerations reveal how the presence of the proα2(I) chain is uniquely adapted to the assembly of heterotrimers of the form [proα1(I)]$_2$proα2(I). First, [proα2(I)]$_3$ homotrimers would be unstable due to repulsion between Lys129 and Lys247 (Fig. 3). This probably explains why inter-chain disulfide bonds between proα2(I) chains do not form even when Ser50 is mutated back to cysteine[36]. In addition, since there are no built-in interaction partners for Lys129 and Lys247 in the proα2(I) chain, these can only interact with residues in the proα1(I) chain. This and the absence of Cys50 would destabilize heterotrimers of the form proα1(I)[proα2(I)]$_2$. Only in the [proα1(I)]$_2$proα2(I) heterotrimer are C-propeptide interactions possible for both Lys129 and Lys247 in the proα2(I) chain, since each of these lysines must interact with a different proα1(I) chain. The importance of the C-terminal residue Lys247 in heterotrimerization is also illustrated by the earlier observation that deletion of the last 10 residues in the proα2(I) chain prevents its incorporation into heterotrimers[37]. Therefore, the sequences of the proα1(I) and proα2(I) C-propeptide chains are perfectly adapted to very specific interactions during the initiation of heterotrimeric procollagen assembly.

In terms of thermal stability, by CD homo-CPI was more resistant to heating than hetero-CPI, as also found when preparing samples for SDS-PAGE (see 'Methods' section). In contrast, the melting profile of CPIII was more similar to that of hetero-CPI than homo-CPI, despite having three inter-chain disulfide bonds like homo-CPI. As shown for CPIII however, this melting transition was irreversible. Thus equilibrium thermodynamics do not apply and no conclusions can be made about differences in thermodynamic stability (that is, Gibbs free energy) between the different types of trimer[38]. This problem is compounded by the presence of inter-chain disulfide bonds which render trimerization an irreversible process. In order to determine whether heterotrimer assembly is energetically favoured compared to homotrimer assembly, measurements are

required of the interactions between monomers during the course of trimerization. Such experiments were possible in the case of the basement membrane collagen IV where the C-terminal NC1 domain, though structurally unrelated to the fibrillar procollagen C-propeptides, also controls homo- and heterotrimerization, but in the absence of inter-chain disulfide bonds[39]. By surface plasmon resonance, it was concluded that $[\alpha1(IV)_2]\alpha2(IV)$ heterotrimers of collagen IV NC1 were more stable than $\alpha1(IV)_3$ homotrimers. Since on average there appear to be more interactions per $\alpha1(I):\alpha2(I)$ interface (Fig. 3c,d) than for the $\alpha1(I):\alpha1(I)$ interface (Fig. 3b), it is tempting to speculate that the same is true for procollagen I C-propeptides. To test this, however, would require mutation of all the inter-chain disulfide bonding cysteines, with unknown effects on subsequent folding and secretion. Such a study is beyond the scope of the work described here.

The role of the triple-helical region in specifying homo- versus heterotrimerization of procollagen is unknown. Mature collagen I homotrimers (that is, following propeptide removal) extracted from *oim* mice (which do not have an $\alpha2(I)$ chain) appear to be more resistant to thermal denaturation than heterotrimers from wild-type mice[24,40], as also observed when comparing recombinant homo- and heterotrimeric collagen I expressed in insect cells[41]. Again however, these data do not reflect differences in thermodynamic stability, but in this case the fact that denaturation of homotrimers is about 100 times slower than for heterotrimers[24]. On the other hand, there is a one-residue stagger between adjacent chains in the triple-helical region, unlike in the coiled-coil stalk region of the C-propeptide where chains are in register. While the structures of the short C-telopeptide regions (separating the end of the triple-helix and the start of the C-propeptide) are unknown, the fact that the C-telopeptide of the $\alpha2(I)$ chain is shorter than that of the $\alpha1(I)$ chain could probably accommodate this transition more easily than for a homotrimer, especially if the order of the chains is $\alpha1\alpha1\alpha2$ as is currently thought[42]. Finally, there may be specific chaperones that ensure heterotrimer assembly[6] or the heterotrimer:homotrimer ratio could be controlled simply at the level of transcription of the *COL1A2* gene[43].

Short synthetic collagen-like peptides have been designed that spontaneously assemble with high selectivity into heterotrimers containing two identical chains and a third distinct chain (AAB arrangement)[44,45]. These peptides exploit the ability of charge–pair interactions, particularly Lys-Asp, to form strong inter-chain interactions[46]. Therefore, the collagen triple helix has the ability to form heterotrimers in the absence of C-propeptides. Indeed, recombinant well-folded heterotrimeric procollagen I has been expressed in yeast cells (co-transfected with prolyl hydroxylase) where the C-propeptides of both the $\alpha1(I)$ and $\alpha2(I)$ chains were replaced by the foldon trimerization sequence from bacteriophage T4 fibritin[47]. This shows that full-length collagen chains are capable of forming heterotrimers when trimerization is initiated non-specifically, albeit that different forms of heterotrimers (AAB, ABA, BAA) may be present. This raises the question of the precise role of the procollagen C-propeptides *in vivo*. The answer must be that in a cell that is synthesizing multiple collagen types, each type of procollagen chain should have its own specific trimerization domain, thereby avoiding the production of unwanted chimeric molecules. Thus, while different kinds of trimerization domain can be used to favour collagen homo- and heterotrimerization *in vitro*, *in vivo* there is a need for further specificity in the assembly process, as provided by the procollagen C-propeptides. A related but distinct scenario has recently been described for the non-fibrillar collagen IX, a heterotrimer of three non-identical chains, where three non-collagenous NC2 domains assemble to make a template that specifies chain association in a neighbouring triple-helix[48].

Finally, we note that the critical residues located in the long stretch of the CRS for CPIII-$\alpha1$, CPI-$\alpha1$ and CPI-$\alpha2$ are all found in the 6/7 residue sequence that is characteristic of the chordate lineage[49–51] (Supplementary Fig. 7). This supports a key role for this sequence in the evolution of the fibrillar collagens. As for the C-terminal lysine which plays an important role in heterotrimer formation as described here, in the human fibrillar procollagens this occurs only in the proα2(I) chain. It is therefore of interest to see when this feature first appeared during evolution. Though the COLFI domain can been traced as far back as the choanoflagellates[52,53], in these species the C-terminal sequence is often truncated and the final cysteine residue is lacking. A C-terminal lysine first appears in several species of arthropods (including insects and crustacea) as well as in the primitive chordates ascidiacea (sea squirts; Supplementary Fig. 7). While the CRS insert has been reported in one ascidian collagen[51], it does not appear in arthropods. Thus the C-terminal lysine precedes the appearance of the CRS insert suggesting that heterotrimer formation may be a relatively early event. In addition, even larger gaps are found in some COLFI domains within the relatively variable $\sim30$ residue region preceding the CRS (Supplementary Fig. 7), such as in the human collagens XI ($\alpha2$ chain), XXIV and XXVII. We note that these gaps correspond to a large loop on the external face of the petal region that is not involved in inter-chain interactions (Supplementary Fig. 8; Supplementary Movie 5), suggesting that its absence would have little effect on trimerization.

In conclusion, here we show how differences in the inter-chain interface involving a relatively small number of residues can account for the specificity of assembly of CPIII and homo-CPI. In addition, we have identified residues in the C-propeptide region of the proα2(I) chain that are uniquely adapted to forming heterotrimers with the C-propeptides of two proα1(I) chains. However, the precise mechanisms that control the ratio of heterotrimers to homotrimers, especially *in vivo*, remain to be elucidated.

## Methods

**Protein expression and purification.** DNA encoding the C-propeptide regions of the human procollagen α1(I) and α2(I) chains, denoted by C1A1 and C1A2, respectively, was amplified by PCR from baculovirus transfer vectors encoding the full-length procollagen αI(I) and α2(I) chains[41] and cloned into pVL1392. For the α1(I) C-propeptide, constructs with or without an N-terminal His_6 tag were generated; for the α2(I) C-propeptide, only an N-terminal His_6 tagged construct was generated. The single-N-glycosylation site in each chain was removed by the mutations Asn147Gln and Asn148Gln in α1(I) and α2(I), respectively (numbering starts at the first residue of the C-propeptide, see Supplementary Fig. 2). The C propeptide constructs were transferred by PCR from pVL1392 into the mammalian expression vector pHLsec[54] using the following primers:

C1A1 forward: 5′-GATCACCGGTGATGATGCCAATGTGGTTC-3′
C1A1His/C1A2His forward: 5′-GATCACCGGTCATCATCATCATCATC ATG-3′
C1A1/C1A1His reverse: 5′-GTCAGTCGACTTACAGGAAGCAGACAGG-3′
C1A2His reverse: 5′-GTCACTCGAGTTATTTGAAACAGACTGGGCC-3′.

PCR fragments were then inserted into the *AgeI/XhoI* sites of pHLsec either directly (C1A2His; following digestion with *AgeI/XhoI*) or indirectly (C1A1 and C1A1His; via blunt end cloning in pJET then digestion with *AgeI/SalI*). To introduce a sequence encoding a TEV protease cleavage site in C1A2His between the His_6 tag and the C-propeptide, two rounds of PCR were carried out using the following primers:

Forward 1: 5′-GAAAACCTGTATTTTCAGGGCGACCAGCCTCGCTCAGCA CCT-3′
Forward 2: 5′-GTAGCTGAAACCGGTCATCATCATCATCATCATGAAAAC CTGTATTTTCAG GGC-3′
Reverse: 5′-GATACTAGTCTCGAGTTACAGGAAGCAGACAGGGCC-3′

The final PCR fragment was then inserted into the *AgeI/XhoI* sites of pHLsec using the In-Fusion HD cloning kit (Clontech) according to the manufacturer's instructions. Briefly, the digested vector and PCR product containing 15 bp complementary ends were mixed in a mass ratio 1:1 then incubated with enzyme

premix as recommended. From the reaction mix, 2.5 µl was then used to transform *Escheria coli* Stellar cells using a standard protocol.

All purified ligation products were checked by double-stranded DNA sequencing (GATC-Biotech). N-terminal sequences of the corresponding protein chains (after removal of the signal peptide) are: α1-ETG<u>DDA</u>; α1His-ETGHHHH HH<u>DDA</u>; α2His-ETGHHHHHH<u>DQP</u>; α2HisTEV-ETGHHHHHHENLYFQ/ G<u>DQP</u> (where / indicates the TEV cleavage site and the sequences underlined correspond to the start of the wild-type protein).

Transient transfection in HEK 293T cells was carried out as described for CPIII (ref. 29), with the following modifications. For large-scale production of homo-CPI, cells at 90% confluence were transfected in Corning Cellstacks (2.5 mg C1A1His DNA for 10-layers in 1 litre of medium) and the medium collected after 72 h. Fresh medium was then added and the cultures continued for a further 48 h. For production of hetero-CPI, cells were co-transfected with C1A1 and C1A2His, in a 2:1 mass ratio, but keeping the total amount of DNA constant. Protein purification was as described[29]. For removal of the His$_6$ tag from α2HisTEV, this step was carried out after the Co$^{2+}$ affinity column by incubation of hetero-CPI with His-tagged TEV protease[55] at a mass ratio of 50:1, overnight at 4 °C in 50 mM Tris-HCl, 0.3 M NaCl, ∼40 mM imidazole, pH 8.0. TEV protease and undigested hetero-CPI were then removed by a further chromatography step on the Co$^{2+}$ column collecting the flow-through fraction. Final purification was carried out using Superdex S200, as described[29]. Typical yields of purified protein for both homo-CPI and hetero-CPI were about 2 mg per litre of culture medium.

**Crystallization and structure determination.** Homo-CPI was first concentrated to 20 mg ml$^{-1}$ in a buffer containing 20 mM Hepes, pH 7.4 and 0.15 M NaCl. Initial crystallization trials were carried out using commercially available sparse matrix screens (Molecular Dimensions Ltd, Hampton Research) and a Mosquito nanolitre pipetting robot (TTP Labtech). Crystals of homo-CPI were grown in hanging drops containing 2 µl protein and 2 µl precipitant over 1 ml of 18% PEG 4000, 0.1 M Tris pH 8.0 using microseeding with crystals obtained from the initial crystallization hits. Glycerol was added to the drop to a final concentration of 17% for cryo-protection prior to flash freezing of crystals in liquid nitrogen. X-ray data collected on beamline PXIII (SLS, Zurich Switzerland), at a wavelength of 1.000 Å, were processed and scaled using programmes from the *XDS* package[56]. Crystals belonged to the space group *P*2$_1$ and diffracted X-rays to 2.2 Å resolution. The crystal structure of homo-CPI was solved by molecular replacement using the programme *PHASER*[57] and the coordinates of CPIII (ref. 28; PDB code 2AEJ) as search model. Refinement of the homo-CPI structure was carried out using restrained and TLS refinement as implemented in the programme *REFMAC* 5 (ref. 58). The programme *COOT*[59] was used for displaying and examining the electron density maps, interactive fitting, optimizing the geometry and also for adding water molecules. The stereochemistry and quality of the homo-CPI structure were analysed using *PROCHECK*[60] (Ramachandran statistics: 87.2% in most favoured regions, 10.7% in additionally allowed regions, 1.7% in generously allowed regions). Interfaces between individual chains in the CPI homotrimer were analysed using the *PDBePISA* server[61].

For hetero-CPI, with or without a His-tag on the α2(I) chain, purified protein was prepared at 11 to 19 mg ml$^{-1}$ in 20 mM Hepes pH 7.4, 0.15 M NaCl. Crystallization trials were set up at 19 °C using the following kits: Crystal screen, JCSG+, Proplex, PACT premier, PEG/Ion, PEGs suite, Stura Footprint. The protein was analysed by dynamic light scattering and by multi-angle laser light scattering and found to be mostly monodisperse with a molecular mass of ∼85 kDa, as expected.

**Mass spectrometry.** Mass spectrometry was performed using a Voyager-DE Pro MALDI-TOF mass spectrometer (AB Sciex) equipped with a nitrogen UV laser (λ = 337 nm, 3 ns pulse). The instrument was operated in the positive-linear mode (mass accuracy: 0.05%) with an accelerating potential of 20 kV. Typically, mass spectra were obtained by accumulation of 600 laser shots for each analysis and processed using Data Explorer 4.0 software (AB Sciex). After desalting with C4 ZipTips (Millipore), samples were mixed with sinapinic acid (saturated solution in 30% acetonitrile and 0.3% trifluoroacetic acid), deposited on the MALDI target and air-dried before analysis. Errors are based on precision as stated by the manufacturer, as well as statistical errors, as a function of signal-to-noise, based on multiple repeats.

**Production and analysis of mutants.** For the C1A1His mutants C1A1His-D126A, C1A1His-D126N, C1A1His-D129A, C1A1His-D129N and C1A1His-E143K, site-directed mutagenesis was carried out by PCR using FideliO DNA polymerase (Ozyme) with C1A1His in the pJET vector as template and the following overlapping oligonucleotides. Each forward primer was individually combined with the C1A1His reverse primer while each reverse primer was combined with the C1A1His forward primer to obtain two PCR products with perfectly complementary sequences flanking the mutation sites. These allowed joining of the two DNA segments with the *Age*I/*Xho*I digested pHLsec by using the In-Fusion HD cloning kit (Clontech) as described previously, thereby incorporating each mutation.

D126A forward: 5′-GGCCAGGGCTCC<u>GCC</u>CCTGCCGATGT-3′

reverse: 5′-ACATCGGCAGGG<u>GC</u>GGAGCCCTGGCC-3′
D126N forward: 5′-GGCCAGGG<u>C</u>TCCAACCCTGCCGATGT-3′
reverse: 5′-ACATCGGCAGG<u>G</u>TTGGAGCCCTGGCC-3′
D129A forward: 5′-TCCGACC<u>CT</u>GCCG<u>CT</u>GTGGCCATCCA-3′
reverse: 5′-TGGATGGCCACA<u>GC</u>GGCAGGGTCGGA-3′
D129N forward: 5′-TCCGACC<u>CT</u>GCCAATGTGGCCATCCA -3′
reverse: 5′-TGGATGGCCACA<u>TT</u>GGCAGGGTCGGA -3′
E143K forward: 5′-CGCCTG<u>A</u>TGTCCACCAAGGCCTCCCAGCAAATC-3′
reverse: 5′-GATTTGCTGGGAGGCC<u>TT</u>GGTGGACATCAGGCG-3′

For the double-mutant C1A1His-D126A/D129A, C1A1His previously mutated to D126A was used as a template with the following overlapping oligonucleotides:
D126A/D129A forward: 5′-TCCGCCCCTGCCG<u>CT</u>GTGGCCATCCA-3′
reverse: 5′–TGGATGGCCACA<u>GC</u>GGCAGGG<u>GC</u>GGA-3′

C1A1His-E143K was then used to generate the non-tagged C1A1-E143K construct using the C1A1 forward and reverse primers.

For the C1A2 mutants, C1A2His-R45A, C1A2His-K129A, C1A2His-E130A and C1A2His-K129A/E130A were generated by two-step PCR as described for the C1A1His mutants but using pHLsec-C1A2His as template, the C1A2His forward and reverse primers, and the following oligonucleotide sequences.
R45A forward: 5′-CGCACATGCGCT<u>G</u>ACTTGAGACTC-3′
reverse: 5′-GTC<u>AGC</u>GCATGTG<u>C</u>GAGCTGGGTT-3′
K129A forward: 5′-GTGACTTCCGC<u>G</u>GAAATGGCTACC-3′
reverse: 5′-TTCCGCGGAAGTCA<u>CT</u>CCTTCTAC-3′
E130A forward: 5′-ACTTCCAAGG<u>C</u>CAATGGCTACCCAA-3′
reverse: 5′-CATTG<u>C</u>CTTGGAAGTCACTCCTTC-3′
K129A/E130A forward: 5′-GTGACTTCCGC<u>G</u>GCAATGGCTACC-3′
reverse: 5′-CATTG<u>CC</u>GCGGAAGTCACT<u>C</u>CTTC-3′

The double-mutant C1A2His-R45A/K129A was generated using the same double-PCR steps with the K129A forward and reverse primers but on the C1A2His-R45A mutant construct. The single-mutant C1A2His-K247A was generated by a single PCR step using a flanking reverse primer and the C1A1His/C1A2His forward primer:
K247A reverse: 5′–GATACTAGTCTCGAGTTATGCGAAACAGACTGGGCC-3′

Finally, to generate the mutations C1A2His-K129A/K247A, C1A2His-R45A/K247A, C1A2His-R45A/K129A/K247A, C1A2His-K129A/E130A/K247A and C1A2His-R45A/K129A/E130A/K247A, the mutants C1A2His-K129A, C1A2His-R45A, C1A2His-R45A/K129A, C1A2His-K129A/E130A and C1A2His-R45A/K129A/E130A were amplified using the C1A2His forward primer and the K247A reverse primer from the C1A2His-K247A construct. All mutations were confirmed by double-stranded DNA sequencing (GATC-Biotech).

To determine the effects of the mutations on trimerization, small scale transfections (using C1A1His or C1A2His alone or with a mixture of C1A1 (non-tagged or His-tagged) and C1A2His) were carried out using 6-well culture plates (Corning), containing 293T cells at 90% confluence, with 4 µg total DNA per well, and the medium collected after 72 h. The medium was then centrifuged at 10,000 g for 15 min at 4 °C, the supernatant collected and protease inhibitors added before storage at −30 °C.

**Electrophoresis and western blotting.** SDS-PAGE was carried out using standard protocols. Homo-CPI was found to be relatively resistant to denaturation in non-reducing conditions, often migrating as a double band consisting of fully and partially denatured protein. To avoid this, it was necessary to denature the samples for 3 min at 100 °C in a final concentration, after addition of sample buffer, of 2% SDS. For analysis of the effects of the mutations on trimerization, 40 µl samples from each 6-well plate culture supernatant were first analysed by SDS-PAGE (4–20% acrylamide gradient gels for non-reducing conditions, 10% acrylamide gels for reducing conditions), after addition of 10 µl 5× sample buffer containing 10% SDS, followed by electrotransfer to Immobilon-P (Millipore) membranes. For western blotting, the first antibody was mouse anti-His-tag (R&D Systems ref. MAB050, dilution 1:5,000) and the second antibody was horseradish peroxidase coupled anti-mouse (Cell Signaling ref. 7076, dilution 1:3,000). For two-dimensional SDS-PAGE, standard 10% acrylamide gels were run in the first dimension in non-reducing conditions. Individual tracks were then cut out (without staining), then reduced (15 min, room temperature) in 100 mM dithiothreitol and alkylated (15 min, room temperature) in 100 mM iodoacetamide, both dissolved in stacking buffer. Individual tracks were then placed horizontally on the top of 10% acrylamide gels (one track per gel), held in place with 0.5% agarose dissolved in stacking buffer and containing 0.1 mg ml$^{-1}$ bromophenol blue, and then run in the second dimension (reducing conditions). Gels were then processed for western blotting as above.

**Circular dichroism.** Far ultraviolet (195–260 nm) CD measurements were carried out using thermostated 0.2 mm path length quartz cells in an Applied Photophysics Chirascan instrument (Protein Science Facility; SFR Biosciences UMS3444/US8). Proteins (0.2–0.7 mg ml$^{-1}$) were analysed at 25 °C in 20 mM Tris-HCl pH 7.4, either with (homo-CPI, hetero-CPI) or without (CPIII) 0.15 M NaCl. Though the absence of NaCl permitted data to be measured at lower wavelengths (with no effect on the spectrum for CPIII), 0.15 M NaCl was included in the buffer for both forms of CPI as homo-CPI (but not hetero-CPI) was found to have reduced solubility without NaCl. Spectra were measured in duplicate using a wavelength

increment of 1 nm, integration time 16 s and bandpass 1 nm. Protein concentrations were determined by Nanodrop. To follow the temperature dependence, CPIII solutions were heated from 25 to 85 °C, in steps of 5 °C (5 min. per step), and then cooled back down to 25 °C. Precise temperatures were measured using a thermocouple probe. At each temperature, spectra were measured (in duplicate) from 190 to 260 nm with wavelength increment 1 nm, integration time 4 s and bandwidth 1 nm. Temperature scans were also carried out at a single wavelength of 208 nm, heating the samples from 25 to 80 °C at 1 °C min⁻¹, and the CD signal measured at 1 °C intervals using a bandpass of 2 nm and integration time 16 s. Observed ellipticities were converted to mean residues molar ellipticities by taking into account mean residue weight, cell path length and protein concentration. First derivatives $(d[\theta]/dt)$ of the curves of mean residues molar ellipticities (at 208 nm) versus temperature were calculated then smoothed using the Savitzky–Golay algorithm.

**Small angle X-ray scattering.** SAXS data were collected at the Diamond Light Source, UK on beamline B21 using the BIOSAXS sample-changing robot. Data were collected at 15 °C in the q range 0.015–0.4 Å⁻¹ with the Pilatus 2M detector (3 min/180 frames per sample). Samples of hetero-CPI consisting of a His-tagged α2(I) chain and non-tagged α1(I) chains were analysed at different protein concentrations in the range 2 to 17 mg ml⁻¹, in 20 mM Hepes pH 7.4, 150 mM NaCl. As there was no evidence of concentration-dependent changes in the scattering profile, data collected for the highest protein concentration were then processed using the programmes *PRIMUS*[62] and *GNOM*[63]. For comparison, the theoretical scattering curve for homo-CPI was calculated from the crystal structure using the *WAXSiS* server[64] and fitted to the experimental scattering curve of hetero-CPI.

**Figure rendering and homology modelling.** Sequence alignment was carried out using the programme *MUSCLE*[65] or the NPS@: Network Protein Sequence Analysis server[66] using sequence data from the UniProt[67] and SMART[68] databases. Structure-based alignment was performed and rendered using the programmes *ESPript 3.0* and *ENDscript 2.0* (ref. 69). Protein structures were drawn using the programme *PyMOL* (Version 1.4.1, Schrödinger, LLC). For modelling of hetero-CPI, all residues in the two interface regions of one of the chains in homo-CPI were first mutated manually to the corresponding residues in the α2(I) chain (Supplementary Fig. 2), using the 'mutagenesis' function in *PyMOL*, then the overall model was energy minimized with *PHENIX*[70].

**Data availability.** The coordinates and structure factors of homo-CPI have been deposited in the Protein Data Bank (http://www.rcsb.org/pdb) with accession number 5K31. Any additional data not shown in this published article or in its Supplementary Information are available from the corresponding authors on reasonable request.

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

## Acknowledgements

This research was funded by the French National Research Agency (project 'TOLLREG'), the European Community's Seventh Framework Programme (FP7/2007–2013; BioStruct-X, grant agreement N°283570), the CNRS, the Université Claude Bernard Lyon 1 and the Région Rhone-Alpes (Ph.D. scholarship to L.C.). Technical support from staff on beamlines B21 (Diamond Light Source), PXIII (Swiss Light Source), MX and FIP (both European Synchrotron Radiation Facility) is gratefully acknowledged. We are also grateful for assistance from A. Lipski (UMR 5086) and from V. Gueguen-Chaignon, I. Zanella-Cléon, and R. Montserret of the Protein Science Facility of SFR Biosciences Lyon (UMS3444/US8).

## Author contributions

U.S. and N.A. carried out the crystallization, structure determination and crystal structure analysis; N.M. cloned the initial constructs; L.C., R.-N.G., S.V.-L.G and D.J.S.H. produced and analysed the site-directed mutants; R.-N.G. and D.J.S.H. carried out the circular dichroism analysis; F.D. carried out the mass spectrometry; J.M. and P.K. supplied the baculovirus constructs; C.M. provided valuable suggestions; D.J.S.H. produced the proteins and carried out the SAXS analysis; D.J.S.H. wrote the paper.

## Additional information

**Competing financial interests:** The authors declare no competing financial interests.

**Publisher's note**: 

