## [Peer Review File · Nature Communications]

Reviewers' Comments:

Reviewer #1 (Remarks to the Author):

The topics of chain selection and the structure of the C-propeptides are important in the fields of collagen, protein synthesis, and self-assembly, and this well-written manuscript presents solid results and original thinking that will impact the directions of these fields. This manuscript reports the high resolution crystal structure of the homotrimer C-terminal propeptide of the $\alpha 1$ chain of type I collagen, which is a very significant accomplishment. This structure is compared with the similar structure of the homotrimer $\alpha 1$ type III C-propeptide previously published by these authors. The differences between the interaction interfaces of neighboring chains suggests how the $\alpha 1$ (I) and $\alpha 1$ (III) chains can both be made in the same cell while preserving their appropriate trimer composition. Type I collagen normally is found as a heterotrimer, composed of two $\alpha 1$ (I) chains and one $\alpha 2$ (I) chain, and the authors produced such C-propeptide heterotrimers but could not obtain good crystals. Since CD spectroscopy and SAXS suggest the low resolution structure of the hetero C-propeptide is similar to that of the homotrimer C-propeptide, modeling was carried out for the heterotrimer and used as a basis for predicting interactions at the interfaces between chains. These predictions were tested with mutagenesis studies that confirmed the importance of unique amino acids in the $\alpha 2$ (I) for interface interactions. For both homotrimers and heterotrimers, the mutation studies suggest there are "backup" interactions to explain why single mutations do not have any effect, while double mutations (or more) do. Overall, this is an impressive contribution which is highly suitable for publication in Nature Communications. However, there are some suggestions that could improve this manuscript.

1. The title should be changed to better reflect the comparison of type I and III homotrimer propeptides as well as the heterotrimers (e.g. Collagen Chain selection in Cells). The question posed in the title is not really answered. Homotrimers of the $\alpha 1$ (I) chain are found in many important physiological and pathological conditions, and the importance of such homotrimers and their differences from type III homotrimers should be given emphasis.
2. The authors should consider mention of the chain selection for type IV collagen, where the crystal structure of the heterotrimer NC1 domain with two $\alpha 1$ (IV) and one $\alpha 2$ (IV) chain was solved.

3. What are the implications of these "backup" interactions? Is this really backup or redundancy or could this finding relate to some dynamic structure, or ensemble of interactions, where perhaps the crystal structure has just captured one of the interactions? In the Discussion, it sometimes sounds as though the C-terminal Lys of the $\alpha 2$ chain is essential for heterotrimers, but changing this residue to Ala did not have any effect; it is only affected in combination with changes in other residues. That seems to disagree with the "key role" of the C-terminus or this residue. The R45A is the only single mutation that reduces heterotrimer formation so that must be key. The mutated double R45A/K247A further reduces heterotrimer formation.
4. It would be helpful to put all the evolutionary implications in one place (and be clear on when heterotrimers and type III collagen arise), or to eliminate this and focus on mechanistic aspects.
5. It is surprising that the homotrimer is more stable than the heterotrimer C-propeptide, suggesting the interactions in the modeled heterotrimer do not contribute much stability. Could the authors explain more fully what a kinetic mechanism would look like?
6. The publication ends without a concluding paragraph that summarizes the important aspects of the paper.

Minor points:

Page 4, line 153. I believe it should read "Supplementary Fig. 1" not Supplementary Fig. 2
Supplementary Fig. 2. It would be nice to have Helix 4 pointed out, with an arrow and label.
Also would be useful to have some indication of residues mentioned e.g. Arg45. It appears that for CPIII that the $\alpha 4$ helix is labeled as a second $\alpha 3$. What is the little bracket below the second $\alpha 3$?

Supplementary Fig. 5. It was difficult to follow some of the residues pointed out in the text. It didn't help that COL1A2, as well as COL1A1 and COL1A3 are buried in the middle of the other genes. And the numbering system does not follow the numbers used in the text or in Supplementary fig. 2, e.g. couldn't find Arg45 in the COL1A2 chain here.

Reviewed by Barbara Brodsky and Barbara D. Smith

Reviewer #2 (Remarks to the Author):

The data presented in this study confirms the basic molecular principles for collagen C-propeptide chain selection (already inferred from the CPIII structures by the same group) and shows why mixed-type fibrillar collagens (e.g. I/III heterotrimers) cannot occur. The data explains convincingly why the [2:1] $\alpha 1/\alpha 2$ CPI heterotrimer can form, and why other combinations are not observed ($\alpha 2$ homotrimer, [1:2] $\alpha 1/\alpha 2$ heterotrimer). Thus, it is a very

important contribution to understanding the folding of the most important collagen type and in my opinion deserves definitely publication in Nature Communications.

However, the study does not provide a complete answer to the fundamental question as formulated in the proposed title, as $\alpha 1$ homotrimers are (somewhat surprisingly) more stable than the [2:1] heterotrimer. This we still don't know for sure why collagen I is "normally" a [2:1] heterotrimer over a more stable (thermodynamically) $\alpha 1$ homotrimer. I would suggest tweaking the title slightly to reflect this fact. Something like "How does collagen I form heterotrimers", or "Molecular basis for heterotrimer chain selection in type I collagen" would be a more realistic description of the findings reported here.

It is a shame that no suitable crystals were obtained for the CPI heterotrimer. Possibly there is unwanted formation of $\alpha 1$ homotrimers together with the desired heterotrimers on the crystallization drops (given the unfavourable thermodynamic balance between homotrimers and heterotrimers reported here). Such heterogeneous sample would probably prevent formation of diffraction-quality crystals. The gels shown in Figure 4 for the [2:1] co-expressed mixtures of $\alpha 1$ and $\alpha 2$ chains would be consistent with that: some $\alpha 2$ chain is detected as monomer, indicating that a fraction of $\alpha 1$ chain is "lost" in homotrimers, leaving a corresponding fraction of $\alpha 2$ without partners to engage in trimer formation.

In their discussion, the authors could elaborate on possible avenues for displacing the equilibrium towards heterotrimer formation, for instance by engineering specific mutations to weaken the $\alpha 1$ homotrimer structure. Nevertheless, in the absence of the actual crystal structure, the mutagenesis data combined with/mapped on the homotrimer CPI structure provide a convincing explanation of the mechanisms of heterotrimer selection, and also why there are no homotrimers of $\alpha 2$ chain (nor heterotrimers with two copies of the $\alpha 2$ chain).

Yet, we are not really any wiser about why collagen I is physiologically a heterotrimer, hence the title of the paper could be somewhat misleading, as discussed above. It is likely that the in vivo prevalence of heterotrimer is the result of the combined stability of CP plus triple helical domains. The authors should discuss this possibility when they mention the heterotrimer formation of peptides and engineered triple helical domains with bacteriophage trimerization foldon domains.

Another important issue that the authors should include in their discussion is the difference in chain stagger/register between the CP trimer and the triple helical domain (from no-stagger in the CP trimeric α -helical coiled-coil to the one-residue stagger in the triple helical domain). The differences in length in the intervening sequence between the end of the triple helical domain and the beginning of the CP domain for $\alpha 1$ and $\alpha 2$ chains will probably accommodate that transition. The particular structure in that transition zone could result in a particular assembly of the full-

length heterotrimer that is favoured over the $\alpha 1$ homotrimer.

Minor issues/suggested changes

Main text

P4 lines 159-60. For the benefit of readers I think that methodological details should be given as well on the failed attempts at crystallizing the hetero-CPI, i.e. what type of sample was used (His-tag present/removed by TEV digestion), what levels of protein concentration were used, was the heterotrimer purified first using size exclusion, was there any light-scattering analysis performed? If space is an issue these details can be easily provided as supplementary material.

P4 lines 169-70. I think that the shape of the Kratky plot shown indicates a well-folded protein (globular macromolecule following Porod's law), but no inferences can be made about it being single-domain or multidomain. Nevertheless, an appropriate reference (probably from Robert Rambo's lab) on interpretation of SAXS plots needs to be added here. Same applies to caption to Supplementary Figure 4.

P12 line 506-07, "non-reducing conditions" stated twice on the same sentence.

Figure 3. I would suggest preparing a supplementary movie where the different interactions highlighted in this figure are shown for the homotrimers CPIII, CPI and then the heterotrimer model of CPI. Ideally each of these panels should be shown as stereofigures (for example to fully appreciate details such as the change in conformation of Arg42 side chain), but this has become impractical nowadays due to space considerations. A short movie, relatively easy to produce using PyMol or similar software, would be a very useful addition to the paper.

Figure 4. The caption needs to explain more clearly the meaning of A1H and A2H labels as those for his-tag carrying chains, otherwise they seem to suggest mutation of an alanine in position 1 or 2 to histidine. I think the caption should also state that the 31 kDa bands in "A1H:A2H" controls are monomers of $\alpha 2(I)$ chain.

Figure 4. I would recommend adding as supplementary material a cartoon representation of the type shown in panel (C) for each of the trimers being discussed in this paper (homo-CPIII, homo-CPI, and hetero-CPI, for comparison). This supplementary figure could be made bigger and additional details on the interactions, mutations, etc could be mapped onto it.

Supplementary Table 1, S-S bridges between Cys47 and Cys64 are listed erroneously as hydrogen bonds in all three interfaces of homo-CPI and CPIII. To simplify the table, the hydrogen bonds between charged side chains could be listed only as salt bridges rather than twice (as hydrogen bonds and salt bridges), as it is always questionable to fix a cutoff to classify

one of these as hydrogen bond (3.25 Å? 3.5 Å?). The actual distance values are given anyway for the benefit of the reader.

Supplementary Figure 2, poor caption. (A) depicts not only secondary structure elements but also the (structure-)based alignment of the different CP sequences. Several visual elements in this alignment (A) are unexplained, such as the dots on top of some residues (these are presumably 10-residue signposts, do they apply only to the top sequence?). Also, the colour-coding for sequence conservation in (A) is not described, and then its mention in (B) makes this caption confusing. Missing elements are: red highlight (used for identical residues), red type (used for conservative changes, e.g. ED, VIL, RK, etc), gray type (no conservation), blue boxes/outlines (unknown meaning), “well”-like drawing underneath $\alpha 3$ helix (unknown meaning). The position of the glycosylation site mentioned in the caption could be marked by a suitable symbol (e.g. star). The caption could help the reader with the numbering of Cys residues by stating their actual colours and/or that these numbers are placed under the sequences.

Caption to (B) is confusing, as many people would actually describe (A) as a structural alignment. I would suggest starting the (B) caption (B) as “Ribbon diagram/3D representation of...” or something along those lines. The “width of the line” term is very confusing. Do authors mean the width of the ribbon used for structural representation? I think this figure could be best served using a simpler $C\alpha$ trace representation of the aligned structures (maintaining a suitable colour-coding representation to indicate sequence conservation). Disulphide bridges are shown but not mentioned in the caption to (B). Could use the same colour-coded numbers used in (A) to label these positions on the 3D structure.

Supplementary Figure 3: define MRW, if mean residue ellipticity, shouldn't it be MRE? I think that “first derivative” is clearer than “first differential”. A non-mathematically proficient reader would appreciate being told that the maxima of these diagrams represent the midpoint of the CD thermal transition curves and that these midpoints are normally used as a quantitative measure of thermal stability. The spectra look saturated at low wavelengths. With the dominance of an α -helical signal we would expect a pronounced maximum of positive ellipticity between 190 and 200 nm. It seems that the spectrum goes to zero because the detector becomes saturated due to the protein concentration (0.2-0.7 mg/ml) and a particularly CD-unfriendly buffer (0.15 M NaCl). It might be preferable to show data only up to the wavelength where there is no saturation (probably 205 nm).

Supplementary Figure 4. (A,B) Authors should compare the experimental value for radius of gyration and maximum dimensions with those calculated from the crystal structure coordinates. (C) I think that the shape of the Kratky plot shown indicates a well-folded protein (globular macromolecule following Porod's law), but no inferences can be made about it being single-domain or multidomain. Nevertheless, an appropriate reference (probably from Robert Rambo's

lab) on interpretation of SAXS plots should be added here. (D) Also a reference for the CRYSOLOG software should be added, and a figure of goodness of fit (typically chi-square) should be given.

Other than these minor suggestions I find the data presented in this paper is compelling and solid, the interpretation convincing, and the science exciting. I strongly recommend publication of this manuscript in Nature Communications.

Dr Jordi Bella
University of Manchester

Reviewer #3 (Remarks to the Author):

This manuscript is a great milestone in collagen biology, as well as in such areas as cancer biology, fibrosis, developmental biology and a number of diseases. It will be of great interest for a wide audience. For the first time it provides (1) atomic details of assembly of a homo-trimeric form and (2) detailed investigation of critical amino acid residues involved in assembly of a prevalent hetero-trimeric form of the most abundant extracellular protein, collagen type I. The study is focused on the C-propeptide domain, which is critically involved in chain selection, trimerization and initiation of the folding of the triple helical portion of the molecule. Mutations within this domain lead to severe abnormalities in bone, skin, tendon, ligaments etc. Some of these mutations were explained based on the authors' present findings. Structural and mutational analysis of this domain provides a solid background for addressing health related issues in hereditary diseases (osteogenesis imperfecta, hearing loss, dentinogenesis imperfecta, Ehlers-Danlos syndrome, etc) and in cancer, osteoarthritis, osteoporosis, fibrosis. Authors also tried to get a crystal structure of the hetero-trimeric C-propeptide domain of collagen I, but it was not possible at this time and should not hamper publication of their significant and highly influential findings. This manuscript might be one of the most significant contributions in collagen biology (and beyond!) in this year. It is well-written, results are clear and thoughtfully discussed, except some confusions described below that I believe could be easily addressed by the authors.

Major points:

- (1) Some clarification is needed for cell transfections with 2:1 ratio of a1:a2 constructs. Are there any solid backgrounds that multiple copies of plasmids enter a single cell nucleus during transient transfection in order to provide correct ratio for transcribed chains? Reviewer's own experience (although very limited) with transient transfections shows that even for a single plasmid the efficiency is never 100%. Unless a hetero-trimer assembles outside the cell (which I guess is not the way collagen molecules assemble) I have hard time to understand how your hetero-trimeric expression system works.
- (2) I found results and discussion on stability of two forms, homo-CPI and hetero-CPI, very

confusing. The authors claim in Results section “the two forms differed in stability, with homo-CPI being noticeably more resistant to thermal denaturation than hetero-CPI” and then in Discussion “Contrary to what might be expected, the CD data indicate that recombinant homo-CPI is more stable than recombinant hetero-CPI”. Those claims are probably (unless the authors have other data, not included here) based on thermal profiles shown in Supplemental Figure 2B, where homo-CPI demonstrates an apparent melting temperature about 8 degrees higher than hetero-CPI. The measure of stability should be the Gibbs free energy, not an apparent melting temperature. But the problem I see here is much bigger. First, to derive the Gibbs energy from a melting transition it should be reversible and in equilibrium (which was not demonstrated for this case). Second, once three chains are cross-linked by disulfide bridges (as the case for homo- and hetero-CPI), the melting transitions reflect unfolding/refolding without possibility to re-shuffle chains, thus not reflecting the thermodynamics of chain assembly!

(3) Another confusion comes from comparing results for stability of the CPI domain (without triple helix) with published results from refs.26, 27, 28 for the triple-helical portion (without CPI).

(4) And finally authors’ conclusion “Therefore, thermodynamic considerations should favour the production of homotrimeric rather than heterotrimeric procollagen I in normal tissues.” seems to be pre-mature as pointed above.

(5) I also think that assumption that “heterotrimer assembly may be driven by kinetic effects, much like the collagen molecule itself which is kinetically but not thermodynamically stable (ref.29)” is based on misinterpreted reference. Reference 29 reports that isolated triple-helical portion of collagen is not stable at physiological temperature, but unfolds very slow (kinetically trapped). I don’t understand how it relates to assembly of CPI.

Minor points:

(1) MALDI-TOF results versus expected weights. Are differences within error limits or reflect imperfect cleavage of signal peptides, post-translation modifications, some proteolysis or other modifications?

(2) Was Cl⁻ ion also observed in the crystal structure of CPIII? If not then it needs clarification, as coordinating residues (Gln62) are also present in CPIII.

(3) Figure 4. Can authors explain a doublet for A2H monomer in 4Biii and also for A1:A2H(+/- mutations) in 4Bii, although it is a singlet for monomer in 4Aii for A1H:A2H and A1H(mut):A2H? Was there some degradation of the monomeric form in some preps?

(4) Is there a certain reason to name a set of chains forming a trimer of homo-CPI as B, C and F versus more usual A, B and C for CPIII (Suppl. Table 1)?

(5) Supplementary Figure 5. Honey bee entry H9KR99_API-ME1547_1771 is obsolete. Hydra entry UPI0002B4503D467_688 does not exist. House fly entry T1PCG7_MUSDO267_492 seems to be truncated (no signal peptide, too short for fibrillar collagen; a non-coding sequence?). Are their genes encoding COLFI domain in well-studied drosophila genome? If not, are COLFI-containing genes in honey bee and house fly erroneous? What about other

arthropods: crustaceans or arachnidae?

Response to reviewers

We are grateful to the editor and to the reviewers for their interest in this work and for their numerous helpful suggestions. Below are the detailed responses to each reviewer. Corresponding changes in the manuscript and in the Supplementary Information are highlighted in yellow. Also, some of the paragraphs in the Discussion have been re-arranged.

Reviewers' comments:

Reviewer #1 (Remarks to the Author):

The topics of chain selection and the structure of the C-propeptides are important in the fields of collagen, protein synthesis, and self-assembly, and this well-written manuscript presents solid results and original thinking that will impact the directions of these fields. This manuscript reports the high resolution crystal structure of the homotrimer C-terminal propeptide of the $\alpha 1$ chain of type I collagen, which is a very significant accomplishment. This structure is compared with the similar structure of the homotrimer $\alpha 1$ type III C-propeptide previously published by these authors. The differences between the interaction interfaces of neighbouring chains suggests how the $\alpha 1$ (I) and $\alpha 1$ (III) chains can both be made in the same cell while preserving their appropriate trimer composition. Type I collagen normally is found as a heterotrimer, composed of two $\alpha 1$ (I) chains and one $\alpha 2$ (I) chain, and the authors produced such C-propeptide heterotrimers but could not obtain good crystals. Since CD spectroscopy and SAXS suggest the low resolution structure of the hetero C-propeptide is similar to that of the homotrimer C-propeptide, modelling was carried out for the heterotrimer and used as a basis for predicting interactions at the interfaces between chains. These predictions were tested with mutagenesis studies that confirmed the importance of unique amino acids in the $\alpha 2$ (I) for interface interactions. For both homotrimers and heterotrimers, the mutation studies suggest there are "backup" interactions to explain why single mutations do not have any effect, while double mutations (or more) do. Overall, this is an impressive contribution which is highly suitable for publication in Nature Communications.

We thank the reviewer for these very positive remarks.

However, there are some suggestions that could improve this manuscript.

1. The title should be changed to better reflect the comparison of type I and III homotrimer propeptides as well as the heterotrimers (e.g. Collagen Chain selection in Cells). The question posed in the title is not really answered. Homotrimers of the $\alpha 1$ (I) chain are found in many important physiological and pathological conditions, and the importance of such homotrimers and their differences from type III homotrimers should be given emphasis.

We agree with the reviewer that the original title was not appropriate as we still don't know for certain why in vivo heterotrimers are preferred to homotrimers. The title has therefore been changed, as suggested, to "Structural basis of homo- and heterotrimerization of collagen I". Also, the first paragraph has been extended to include the presence of homotrimers in embryonic tissues and also homo- and heterotrimers of collagen V in skin.

2. The authors should consider mention of the chain selection for type IV collagen, where the crystal structure of the heterotrimer NC1 domain with two $\alpha 1$ (IV) and one $\alpha 2$ (IV) chain was solved.

This is an excellent suggestion. Paragraph 4 in the Discussion (page 8) has been revised to include the highly relevant work done by the Hudson lab on collagen IV assembly.

3. What are the implications of these "backup" interactions? Is this really backup or redundancy or could this finding relate to some dynamic structure, or ensemble of interactions, where perhaps the crystal structure has just captured one of the interactions? In the Discussion, it sometimes sounds as though the C-terminal Lys of the $\alpha 2$ chain is essential for heterotrimers, but changing this residue to Ala did not have any effect; it is only affected in combination with changes in other residues. That seems to disagree with the "key role" of the C-terminus or this residue. The R45A is the only single mutation that reduces heterotrimer formation so that must be key. The mutated double R45A/K247A further reduces heterotrimer formation.

The term "backup" has been removed. Redundancy is probably more appropriate and we have extended paragraph 1 in the Discussion to cover the possibility of alternative conformations as raised by this reviewer.

We agree that a "key role" for the C-terminal Lys may be an exaggeration. The word "key" has been replaced by "important". It should be noted however that the readout for trimerization in this work is rather crude, being entirely dependent on the formation of inter-chain disulphide bonds to stabilize the trimer for subsequent analysis by SDS-PAGE and western blotting. It may well be that there are marked differences in the energetics of the inter-chain interactions between wild-type and mutants, but such information is not currently accessible for the reasons given in paragraph 4 of the Discussion (page 8).

4. It would be helpful to put all the evolutionary implications in one place (and be clear on when heterotrimers and type III collagen arise), or to eliminate this and focus on mechanistic aspects.

All the evolutionary aspects have now been combined in a new paragraph at the end of the Discussion, along with new Supplementary Figures 6 and 7.

5. It is surprising that the homotrimer is more stable than the heterotrimer C-propeptide, suggesting the interactions in the modelled heterotrimer do not contribute much stability. Could the authors explain more fully what a kinetic mechanism would look like?

This point was also raised in some detail by Reviewer 3. In the light of these comments, we are now convinced that the CD data provide no information on thermodynamic stability (i.e. Gibbs free energy) as the melting curves are irreversible. A further problem is the formation of inter-chain disulphide bonds which preclude studies of the energetics of the interactions between chains. This is now discussed in detail in paragraph 4 in the Discussion (page 8). Reference to a kinetic mechanism has been removed.

6. The publication ends without a concluding paragraph that summarizes the important aspects of the paper.

A concluding paragraph has now been added.

Minor points:

Page 4, line 153. I believe it should read "Supplementary Fig. 1" not Supplementary Fig. 2

Thank you for pointing out this error which has now been corrected.

Supplementary Fig. 2. It would be nice to have Helix 4 pointed out, with an arrow and label. Also would be useful to have some indication of residues mentioned e.g. Arg45. It appears that for CPIII that the $\alpha 4$ helix is labelled as a second $\alpha 3$. What is the little bracket below the second $\alpha 3$?

Supplementary Figure 2 has been modified as requested, the label for helix 4 has been corrected and the bracket has been removed.

Supplementary Fig. 5. It was difficult to follow some of the residues pointed out in the text. It didn't help that COL1A2, as well as COL1A1 and COL1A3 are buried in the middle of the other genes. And the numbering system does not follow the numbers used in the text or in Supplementary fig. 2, e.g. couldn't find Arg45 in the COL1A2 chain here.

Supplementary Figure 5 (now Supplementary Figure 6) has been totally revised and expanded to include additional sequences from arthropods and ascidians. The numbering now corresponds to the rest of the manuscript and COL1A1 is now at the top.

Reviewer #2 (Remarks to the Author):

The data presented in this study confirms the basic molecular principles for collagen C-propeptide chain selection (already inferred from the CPIII structures by the same group) and shows why mixed-type fibrillar collagens (e.g. I/III heterotrimers) cannot occur. The data explains convincingly why the [2:1] $\alpha 1/\alpha 2$ CPI heterotrimer can form, and why other combinations are not observed ($\alpha 2$ homotrimer, [1:2] $\alpha 1/\alpha 2$ heterotrimer). Thus, it is a very important contribution to understanding the folding of the most important collagen type and in my opinion deserves definitely publication in Nature Communications.

We thank the reviewer for these very positive remarks.

However, the study does not provide a complete answer to the fundamental question as formulated in the proposed title, as $\alpha 1$ homotrimers are (somewhat surprisingly) more stable than the [2:1] heterotrimer. This we still don't know for sure why collagen I is "normally" a [2:1] heterotrimer over a more stable (thermodynamically) $\alpha 1$ homotrimer. I would suggest tweaking the title slightly to reflect this fact. Something like "How does collagen I form heterotrimers", or "Molecular basis for heterotrimer chain selection in type I collagen" would be a more realistic description of the findings reported here.

We agree with the reviewer that the manuscript does not provide a definitive explanation for why heterotrimers are favoured over homotrimers in vivo. The title has therefore been changed as suggested. With regard to differences in stability, we now agree with reviewer 3 that the CD data do not allow any conclusions to be made about thermodynamic stability as the changes are irreversible. This point is now discussed in detail in paragraph 4 of the Discussion (page 8).

It is a shame that no suitable crystals were obtained for the CPI heterotrimer. Possibly there is unwanted formation of $\alpha 1$ homotrimers together with the desired heterotrimers on the crystallization drops (given the unfavourable thermodynamic balance between homotrimers and heterotrimers reported here). Such heterogeneous sample would probably prevent

formation of diffraction-quality crystals. The gels shown in Figure 4 for the [2:1] co-expressed mixtures of $\alpha 1$ and $\alpha 2$ chains would be consistent with that: some $\alpha 2$ chain is detected as monomer, indicating that a fraction of $\alpha 1$ chain is “lost” in homotrimers, leaving a corresponding fraction of $\alpha 2$ without partners to engage in trimer formation.

For the heterotrimer, the His tag was present only on the $\alpha 2(I)$ chain, thus allowing separation of heterotrimers from homotrimers during purification. There are no signs of any free $\alpha 2(I)$ chains in the purified protein, as shown in Supplementary Figure 1. The reviewer is correct that there were free $\alpha 2(I)$ chains in the crude culture medium, as shown in Figure 4, but these were separated from heterotrimers in the gel filtration step during purification.

In their discussion, the authors could elaborate on possible avenues for displacing the equilibrium towards heterotrimer formation, for instance by engineering specific mutations to weaken the $\alpha 1$ homotrimer structure.

It is difficult to imagine ways of displacing the equilibrium towards heterotrimers given that the same inter-chain interaction between $\alpha 1(I)$ chains occurs in both homo- and heterotrimers. Such a strategy would require strengthening the interactions between unlike chains. However, this would require detailed knowledge of the energetics of the interactions, something that would be difficult to obtain as discussed in paragraph 4 of the Discussion (page 8).

Nevertheless, in the absence of the actual crystal structure, the mutagenesis data combined with/mapped on the homotrimer CPI structure provide a convincing explanation of the mechanisms of heterotrimer selection, and also why there are no homotrimers of $\alpha 2$ chain (nor heterotrimers with two copies of the $\alpha 2$ chain).

Thank you.

Yet, we are not really any wiser about why collagen I is physiologically a heterotrimer, hence the title of the paper could be somewhat misleading, as discussed above. It is likely that the in vivo prevalence of heterotrimer is the result of the combined stability of CP plus triple helical domains. The authors should discuss this possibility when they mention the heterotrimer formation of peptides and engineered triple helical domains with bacteriophage trimerization foldon domains.

We thank the reviewer for this suggestion. It has also been reported however that collagen I homotrimers are more stable than heterotrimers, in the absence of the propeptides, as discussed in paragraph 5 of the Discussion (page 8).

Another important issue that the authors should include in their discussion is the difference in chain stagger/register between the CP trimer and the triple helical domain (from no-stagger in the CP trimeric α -helical coiled-coil to the to one-residue stagger in the triple helical domain). The differences in length in the intervening sequence between the end of the triple helical domain and the beginning of the CP domain for $\alpha 1$ and $\alpha 2$ chains will probably accommodate that transition. The particular structure in that transition zone could result in a particular assembly of the full-length heterotrimer that is favoured over the $\alpha 1$ homotrimer.

We thank the reviewer for this excellent suggestion which has also been included in paragraph 5 of the Discussion (page 8).

Minor issues/suggested changes:

Main text

P4 lines 159-60. For the benefit of readers, I think that methodological details should be given as well on the failed attempts at crystallizing the hetero-CPI, i.e. what type of sample was used (His-tag present/removed by TEV digestion), what levels of protein concentration were used, was the heterotrimer purified first using size exclusion, was there any light-scattering analysis performed? If space is an issue these details can be easily provided as supplementary material.

A new paragraph has been added to the “Crystallization, structure determination and modelling” section of the Methods giving details of the concentrations used and the crystallization conditions tested (approximately 1200) for recombinant hetero-CPI, both with and without the His-tag. Samples were analyzed by DLS and MALLS, giving good results, as also described in this new paragraph. Details of the purification steps, including gel filtration, are given in the section “Protein expression and purification”.

For protein expression, it was necessary to use a mammalian expression system, particularly because of the intra- and inter-chain disulphide bonds. This was very labour intensive and resulted in only moderate yields, thus requiring several rounds of expression and purification over the course of 2-3 years, none of which resulted in crystals. We even tried native hetero-CPI from cultured chick embryo tendons (1.8 mg of purified protein from 240 17-day old eggs!) but again no crystals were obtained.

P4 lines 169-70. I think that the shape of the Kratky plot shown indicates a well-folded protein (globular macromolecule following Porod’s law), but no inferences can be made about it being single-domain or multidomain. Nevertheless, an appropriate reference (probably from Robert Rambo’s lab) on interpretation of SAXS plots needs to be added here. Same applies to caption to Supplementary Figure 4.

We have added a reference to Putnam et al, 2007 on the interpretation of SAXS data and also a reference to <https://www-ssrl.slac.stanford.edu/~saxs/analysis/assessment.htm> (see figure below) which justifies the statement that hetero-CPI is a well-folded multi-domain protein.

P12 line 506-07, “non-reducing conditions” stated twice on the same sentence.

Thank you. This has now been corrected.

Figure 3. I would suggest preparing a supplementary movie where the different interactions highlighted in this figure are shown for the homotrimers CPIII, CPI and then the heterotrimer model of CPI. Ideally each of these panels should be shown as stereofigures (for example to fully appreciate details such as the change in conformation of Arg42 side chain), but this has become impractical nowadays due to space considerations. A short movie, relatively easy to produce using PyMol or similar software, would be a very useful addition to the paper.

Supplementary movies have now been added to complement Figure 3 and Supplementary Figure 7.

Figure 4. The caption needs to explain more clearly the meaning of A1H and A2H labels as those for his-tag carrying chains, otherwise they seem to suggest mutation of an alanine in position 1 or 2 to histidine. I think the caption should also state that the 31 kDa bands in “A1H:A2H” controls are monomers of $\alpha 2(I)$ chain.

The figure legend has now been changed as suggested.

Figure 4. I would recommend adding as supplementary material a cartoon representation of the type shown in panel (C) for each of the trimers being discussed in this paper (homo-CPIII, homo-CPI, and hetero-CPI, for comparison). This supplementary figure could be made bigger and additional details on the interactions, mutations, etc could be mapped onto it.

The interactions in the homotrimer are the same as for the $\alpha 1(I):\alpha 1(I)$ interface in the heterotrimer, as shown in Figure 4C. These are also listed in Supplementary Table 1, as are those for CPIII.

Supplementary Table 1, S-S bridges between Cys47 and Cys64 are listed erroneously as hydrogen bonds in all three interfaces of homo-CPI and CPIII. To simplify the table, the hydrogen bonds between charged side chains could be listed only as salt bridges rather than twice (as hydrogen bonds and salt bridges), as it is always questionable to fix a cutoff to classify one of these as hydrogen bond (3.25 Å? 3.5 Å?). The actual distance values are given anyway for the benefit of the reader.

This table has now been modified as suggested.

Supplementary Figure 2, poor caption. (A) depicts not only secondary structure elements but also the (structure-)based alignment of the different CP sequences. Several visual elements in this alignment (A) are unexplained, such as the dots on top of some residues (these are presumably 10-residue signposts, do they apply only to the top sequence?). Also, the colour-coding for sequence conservation in (A) is not described, and then its mention in (B) makes this caption confusing. Missing elements are: red highlight (used for identical residues), red type (used for conservative changes, e.g. ED, VIL, RK, etc), gray type (no conservation), blue boxes/outlines (unknown meaning), “well”-like drawing underneath $\alpha 3$ helix (unknown meaning). The position of the glycosylation site mentioned in the caption could be marked by a suitable symbol (e.g. star). The caption could help the reader with the numbering of Cys residues by stating their actual colours and/or that these numbers are placed under the sequences.

The legend has been entirely re-written to incorporate all of these most helpful suggestions.

Caption to (B) is confusing, as many people would actually describe (A) as a structural alignment. I would suggest starting the (B) caption (B) as “Ribbon diagram/3D representation of...” or something along those lines. The “width of the line” term is very confusing. Do authors mean the width of the ribbon used for structural representation? I think this figure could be best served using a simpler C α trace representation of the aligned structures (maintaining a suitable colour-coding representation to indicate sequence conservation). Disulphide bridges are shown but not mentioned in the caption to (B). Could use the same colour-coded numbers used in (A) to label these positions on the 3D structure.

We prefer to describe part (A) as a sequence alignment since the programme used to prepare this diagram (ESPRIT) simply adds secondary structure information to pre-aligned sequences. On the other hand, part (B) is a true 3D structural alignment done using the programme ENDSCRIPT. We agree with the reviewer that “width of the ribbon” is more appropriate. We prefer this representation to a C α trace. Various features of the structure, including the disulphide bond positions, have now been added.

Supplementary Figure 3: define MRW, if mean residue ellipticity, shouldn't it be MRE? I think that “first derivative” is clearer than “first differential”. A non-mathematically proficient reader would appreciate being told that the maxima of these diagrams represent the midpoint of the CD thermal transition curves and that these midpoints are normally used as a quantitative measure of thermal stability. The spectra look saturated at low wavelengths. With the dominance of an α -helical signal we would expect a pronounced maximum of positive ellipticity between 190 and 200 nm. It seems that the spectrum goes to zero because the detector becomes saturated due to the protein concentration (0.2-0.7 mg/ml) and a particularly CD-unfriendly buffer (0.15 M NaCl). It might be preferable to show data only up to the wavelength where there is no saturation (probably 205 nm).

MRW (mean residue weight) is now defined in the revised legend. The parameter that is shown is the mean residue molar ellipticity (MRME, also defined) which depends on the MRW. The curves in part (C) (formerly B) have been described in more detail, as requested. Given the comments of reviewer 3 however, we have avoided making the link to stability. We have adjusted the lower wavelength limit as suggested. Note that a new part (B) has been added in response to the comments from reviewer 3.

Supplementary Figure 4. (A,B) Authors should compare the experimental value for radius of gyration and maximum dimensions with those calculated from the crystal structure coordinates. (C) I think that the shape of the Kratky plot shown indicates a well-folded protein (globular macromolecule following Porod's law), but no inferences can be made about it being single-domain or multidomain. Nevertheless, an appropriate reference (probably from Robert Rambo's lab) on interpretation of SAXS plots should be added here. (D) Also a reference for the CRY SOL software should be added, and a figure of goodness of fit (typically chi-square) should be given.

(A, B) We have added (page 4) values for the radius of gyration and the maximum dimension calculated from the crystal structure of homo-CPI (C) This point has previously been dealt with above (see “P4 lines 169-70”). (D) We have recalculated the fit to the data for hetero-CPI using the programme WAXSiS and have included a corresponding reference.

Other than these minor suggestions I find the data presented in this paper is compelling and solid, the interpretation convincing, and the science exciting. I strongly recommend publication of this manuscript in Nature Communications.

Thank you.

Reviewer #3 (Remarks to the Author):

This manuscript is a great milestone in collagen biology, as well as in such areas as cancer biology, fibrosis, developmental biology and a number of diseases. It will be of great interest for a wide audience. For the first time it provides (1) atomic details of assembly of a homotrimeric form and (2) detailed investigation of critical amino acid residues involved in assembly of a prevalent hetero-trimeric form of the most abundant extracellular protein, collagen type I. The study is focused on the C-propeptide domain, which is critically involved in chain selection, trimerization and initiation of the folding of the triple helical portion of the molecule. Mutations within this domain lead to severe abnormalities in bone, skin, tendon, ligaments etc. Some of these mutations were explained based on the authors' present findings. Structural and mutational analysis of this domain provides a solid background for addressing health related issues in hereditary diseases (osteogenesis imperfecta, hearing loss, dentinogenesis imperfecta, Ehlers-Danlos syndrome, etc) and in cancer, osteoarthritis, osteoporosis, fibrosis. Authors also tried to get a crystal structure of the hetero-trimeric C-propeptide domain of collagen I, but it was not possible at this time and should not hamper publication of their significant and highly influential findings. This manuscript might be one of the most significant contributions in collagen biology (and beyond!) in this year. It is well-written, results are clear and thoughtfully discussed, except some confusions described below that I believe could be easily addressed by the authors.

We thank the reviewer for these very positive remarks.

Major points:

(1) Some clarification is needed for cell transfections with 2:1 ratio of $\alpha 1:\alpha 2$ constructs. Are there any solid backgrounds that multiple copies of plasmids enter a single cell nucleus during transient transfection in order to provide correct ratio for transcribed chains? Reviewer's own experience (although very limited) with transient transfections shows that even for a single plasmid the efficiency is never 100%. Unless a hetero-trimer assembles outside the cell (which I guess is not the way collagen molecules assemble) I have hard time to understand how your hetero-trimeric expression system works.

It is true that the co-expression system used was not ideal. We also wondered what percentage of cells actually received both DNAs, and what the DNA ratios were in individual cells. Nevertheless, we did manage to get heterotrimers expressed using this system, with the correct ratio of $\alpha 1(I)$ and $\alpha 2(I)$ chains, after purification, as shown in Supplementary Figure 1. Homotrimers of $\alpha 1(I)$ chains were also expressed with the co-transfections, as found when His tags were present on both chains (see new Supplementary Figure 5), but these could be removed during purification when the His tag was present only on the $\alpha 2(I)$ chain. We assume that trimerization occurred within cells, particularly in view of the disulphide bonding that normally takes place within the endoplasmic reticulum. Some free $\alpha 2(I)$ chains were

found in the medium however (Figure 4), perhaps as a result of saturating the system with such large amounts of DNA?

(2) I found results and discussion on stability of two forms, homo-CPI and hetero-CPI, very confusing. The authors claim in Results section “the two forms differed in stability, with homo-CPI being noticeably more resistant to thermal denaturation than hetero-CPI” and then in Discussion “Contrary to what might be expected, the CD data indicate that recombinant homo-CPI is more stable than recombinant hetero-CPI”. Those claims are probably (unless the authors have other data, not included here) based on thermal profiles shown in Supplemental Figure 2B, where homo-CPI demonstrates an apparent melting temperature about 8 degrees higher than hetero-CPI. The measure of stability should be the Gibbs free energy, not an apparent melting temperature. But the problem I see here is much bigger. First, to derive the Gibbs energy from a melting transition it should be reversible and in equilibrium (which was not demonstrated for this case). Second, once three chains are cross-linked by disulfide bridges (as the case for homo- and hetero-CPI), the melting transitions reflect unfolding/refolding without possibility to re-shuffle chains, thus not reflecting the thermodynamics of chain assembly!

We thank the reviewer for raising this very important point. We have added new data in Supplementary Figure 3, for the C-propeptide of procollagen III (CPIII). Surprisingly, the main melting transition for CPIII (which has three inter-chain S-S bonds) is similar to that for hetero-CPI (two inter-chain S-S bonds). We also include CD spectra for CPIII as a function of temperature, which shows that indeed this transition is irreversible. Thus equilibrium thermodynamics does not apply and no conclusions can be made about stability in the thermodynamic sense. New paragraphs have been added in both Results (page 6) and Discussion (page 8) addressing this point in detail.

(3) Another confusion comes from comparing results for stability of the CPI domain (without triple helix) with published results from refs.26, 27, 28 for the triple-helical portion (without CPI).

This point has been clarified in a new paragraph 4 in the Discussion (page 8).

(4) And finally authors’ conclusion “Therefore, thermodynamic considerations should favour the production of homotrimeric rather than heterotrimeric procollagen I in normal tissues.” seems to be pre-mature as pointed above.

We agree with the reviewer. This sentence has been modified in the new paragraph 4 in the Discussion (page 8).

(5) I also think that assumption that “heterotrimer assembly may be driven by kinetic effects, much like the collagen molecule itself which is kinetically but not thermodynamically stable (ref.29)” is based on misinterpreted reference. Reference 29 reports that isolated triple-helical portion of collagen is not stable at physiological temperature, but unfolds very slow (kinetically trapped). I don’t understand how it relates to assembly of CPI.

The discussion of kinetic effects has been removed from the revised version and replaced by the new paragraph 4 in the Discussion (page 8).

Minor points:

(1) MALDI-TOF results versus expected weights. Are differences within error limits or reflect imperfect cleavage of signal peptides, post-translation modifications, some proteolysis or other modifications?

We have re-analyzed the mass spectrometry data and associated errors, with an explanation in the Methods. These values have now been used for the MALDI-TOF data on pages 3 & 4.

(2) Was Cl⁻ ion also observed in the crystal structure of CPIII? If not then it needs clarification, as coordinating residues (Gln62) are also present in CPIII.

No chloride ions were observed in any of the crystal structures of CPIII, despite the similar composition of the buffer in which all the proteins were stocked. Compared to homo-CPI, there is not enough space for chloride in this position in CPIII. As for the chloride observed in homo-CPI, this most probably comes from the buffer. First it was refined as a water molecule, but the Fo-Fc electron density prompted us to suggest that it was chloride according to the density and ligands. Clearly when describing the density as a chloride the refinement was successful. A further issue is that this chloride is also ligated by Arg39 which also is conserved in CPIII, and this has now been included in the Results section (p. 3). We are sorry for this omission in the original manuscript, which nevertheless has no impact on the message which is that this ion does not seem to have an important structural role.

(3) Figure 4. Can authors explain a doublet for A2H monomer in 4Biii and also for A1:A2H(+/-mutations) in 4Bii, although it is a singlet for monomer in 4Aii for A1H:A2H and A1H(mut):A2H? Was there some degradation of the monomeric form in some preps?

We did find multiple bands for A2H monomers in the western blots, but not after purification of the heterotrimer (Supplementary Figure 1, reducing conditions). This and the fact that proteinase inhibitors were added immediately after medium collection suggests that this was not due to proteolysis. Our interpretation is that different forms of A2H were expressed, only one of which ended up in heterotrimers, the others being perhaps variants due to differences in post-translation modifications, adducts involving the free cysteine or even failure to cleave the signal peptide. In 4Aii, for the co-expression, both chains were His tagged, unlike in 4B. We think that the band at 30 kDa is probably A1H rather than A2H, based on data obtained using 2D gels (see the new Supplementary Figure 5). These show that the band corresponding to monomer in non-reducing conditions migrates with A1H rather than A2H in reducing conditions.

(4) Is there a certain reason to name a set of chains forming a trimer of homo-CPI as B, C and F versus more usual A, B and C for CPIII (Suppl. Table 1)?

We agree, it doesn't seem very logical. The problem is that there are two homo-CPI trimers in the asymmetric unit (i.e. 6 chains). Chains were defined by the molecular replacement solution from A to F, with BCF in trimer 1 and ADE in trimer 2. A note to this effect has been added to the legend for Supplementary Table 1.

(5) Supplementary Figure 5. Honey bee entry H9KR99_API-ME1547_1771 is obsolete. Hydra entry UPI0002B4503D467_688 does not exist. House fly entry T1PCG7_MUSDO267_492 seems to be truncated (no signal peptide, too short for fibrillar collagen; a non-coding sequence?). Are their genes encoding COLFI domain in well-studied

drosophila genome? If not, are COLFI-containing genes in honey bee and house fly erroneous? What about other arthropods: crustaceans or arachnidae?

We thank the reviewer for checking these files in such detail. The problem was due to the fact that some of the files in the SMART database are indeed obsolete. These have now been removed. As for the short sequences, presumably these are fragments of longer proteins. To avoid any doubt however, we have completely revised and extended Supplementary Figure 5 (now Supplementary Figure 6) using full-length active sequences from the UniProt database. Data for four species of arthropods are now included, including one crustacean. Surprisingly, there appear to be no fibrillar collagens in drosophila. A new Supplementary Figure 7 has also been added concerning the significance of the gaps in the sequence alignments in new Supplementary Figure 6.

Reviewers' Comments:

Reviewer #1 (Remarks to the Author):

The authors have responded thoughtfully to all points we raised, and the revised manuscript is an outstanding contribution to the field, and should be published in Nature Communications.

Reviewer #2 (Remarks to the Author):

The authors have addressed satisfactorily all the issues raised by the reviewers on the first version of the manuscript. The manuscript has certainly improved during the review process and it is now ready for publication.

Reviewer #3 (Remarks to the Author):

I would like to thank the authors for prompt and detailed responses to the raised questions and concerns. Whereas most of the points were adequately addressed, there are some that still need to be clarified:

1. Methods, section on expression and purification: The authors still state that: "Typical yields of purified protein for both homo-CPI and hetero-CPI were about 2 mg/litre of culture medium." Supp.Fig.5 shows that although the 2:1 stoichiometric ratio of DNA was used for transfection (constructs A1H:A2H) the real purified material is mainly A1H (A1H:A2H is 20-50:1??). The same is expected for A1:A2H system. I totally agree that if you pull down only A2H from A1:A2H system you'll get the pure A1:A2H hetero-trimer with the correct stoichiometry, but the yield should be much less. Supp.Fig.5 also tells me that either co-transfection of both plasmids into the same cell is a rare event or the expression level of the a2 construct is much lower in general. In any case "about the same" yield for both homo- and hetero-CPIs seems unreal.

2. Last paragraph in Results, p.6: Actually, there are two characteristic minima for alpha-helical structures, namely 208 nm and 222 nm. Thus someone would expect contemporary change of CD signals at 208 nm and 222 nm upon heating, which is not the case for the thermal transition presented in Supp. Fig. 3B. If authors would measure and analyze transitions at 222 nm, their apparent temperature dependencies would look very different from those presented in Supp. Fig. 3C. Thus the meaning of the data presented in Supp. Fig. 3C is questionable.

3. Discussion, p.8, paragraph 2: "Mature collagen I homotrimers (i.e. following propeptide

removal) from the skin of oim mice (which do not have an $\alpha 2(I)$ chain) appear to be more stable than heterotrimers from wild-type skin (Refs. 23,39), as also observed comparing recombinant homo- and heterotrimeric collagen I expressed in insect cells (ref. 40). Thus if the triple-helical region does play a role in chain selection, homotrimers might be favoured over heterotrimers.” Once again, the authors misinterpret thermal transition as thermodynamic stability. To make such statement the Gibbs free energies need to be derived and compared for homo- and heterotrimers at physiological conditions (certain buffer, temperature), which is again problematic due to irreversible nature of thermal transitions presented in refs. 23, 39, 40.

4. With the recently published data from Bachinger’s group on a crystal structure of the triple helix – hetero trimerization domain interface of type IX collagen, it would be interesting to analyze if similar organization is possible in fibrillar collagens.

Reviewer #3 (Remarks to the Author):

I would like to thank the authors for prompt and detailed responses to the raised questions and concerns. Whereas most of the points were adequately addressed, there are some that still need to be clarified:

1. Methods, section on expression and purification: The authors still state that: “Typical yields of purified protein for both homo-CPI and hetero-CPI were about 2 mg/litre of culture medium.” Supp.Fig.5 shows that although the 2:1 stoichiometric ratio of DNA was used for transfection (constructs A1H:A2H) the real purified material is mainly A1H (A1H:A2H is 20-50:1??). The same is expected for A1:A2H system. I totally agree that if you pull down only A2H from A1:A2H system you’ll get the pure A1:A2H hetero-trimer with the correct stoichiometry, but the yield should be much less. Supp.Fig.5 also tells me that either co-transfection of both plasmids into the same cell is a rare event or the expression level of the a2 construct is much lower in general. In any case “about the same” yield for both homo- and hetero-CPIs seems unreal.

The yields of purified protein (Supplementary Fig. 1), starting from conditioned culture medium (volumes in parentheses), for the large scale production of homo-CPI (His tagged) and hetero-CPI (His tag only on $\alpha 2$) were as follows:

Homo-CPI:

Batch 1 (2 litres), 4.75 mg Batch 2 (2 litres), 1.98 mg

Hetero-CPI

Batch 1 (4 litres), 8.82 mg Batch 2 (2 litres), 1.22 mg*

*(*contains a TEV cleavage site thus requiring an additional purification step)*

So the yields for both homo-CPI and hetero-CPI were indeed about 2 mg/litre, as stated in the manuscript.

Concerning Supplementary Fig. 5, we have scanned the western blot data to quantify (subject to the limits of this approach) the amounts of $\alpha 1$ and $\alpha 2$ chains in the trimer, resulting in a ratio of 12:1 in terms of chains. However, given that the heterotrimer contains two $\alpha 1$ chains, this is equivalent to $(10/3 =) 3.33$ molecules of homotrimer for every $(3/3 =) 1$ molecule of heterotrimer. So it is indeed surprising that the yields of heterotrimer in the large scale preps were similar to those for homotrimer. We can think of two possible explanations:

(i) One cannot assume that expression of A1H:A2H will be the same as for A1:A2H. It is well known that small changes in constructs used for protein expression can have major effects on yields. So the presence or absence of the His-tag on the $\alpha 1$ chain could be such an example.

(ii) It may also be that co-expression with A2H boosts total expression of A1, resulting in the observed levels of expression of heterotrimer and even higher amounts (unseen) of homotrimer.

Unfortunately, there is no way of checking these hypotheses (even with separate tags on each chain) as the system does not allow purification of both homo-CPI and hetero-CPI from the same prep.

2. Last paragraph in Results, p.6: Actually, there are two characteristic minima for alpha-helical structures, namely 208 nm and 222 nm. Thus someone would expect contemporary change of CD signals at 208 nm and 222 nm upon heating, which is not the case for the thermal transition presented in Supp. Fig. 3B. If authors would measure and analyze transitions at 222 nm, their

apparent temperature dependencies would look very different from those presented in Supp. Fig. 3C. Thus the meaning of the data presented in Supp. Fig. 3C is questionable.

We agree that there are two characteristic minima for alpha-helical structures. The text (page 6) has been modified to make this explicit. But it does not follow that similar changes would be expected at 220 nm, since the CPI structure is mostly beta-sheet and coil. In any case, a wavelength of 208 nm was chosen simply because this was the part of the spectrum where temperature-dependent changes were most pronounced. We in no way wanted to imply that just the α -helical regions were undergoing change.

3. Discussion, p.8, paragraph 2: “Mature collagen I homotrimers (i.e. following propeptide removal) from the skin of oim mice (which do not have an $\alpha 2(I)$ chain) appear to be more stable than heterotrimers from wild-type skin (Refs. 23,39), as also observed comparing recombinant homo- and heterotrimeric collagen I expressed in insect cells (ref. 40). Thus if the triple-helical region does play a role in chain selection, homotrimers might be favoured over heterotrimers.” Once again, the authors misinterpret thermal transition as thermodynamic stability. To make such statement the Gibbs free energies need to be derived and compared for homo- and heterotrimers at physiological conditions (certain buffer, temperature), which is again problematic due to irreversible nature of thermal transitions presented in refs. 23, 39, 40.

We thank the reviewer for raising this point. We have revised the corresponding section accordingly (page 8), making it clear that again no conclusions can be made about thermodynamic stability.

4. With the recently published data from Bachinger’s group on a crystal structure of the triple helix – hetero trimerization domain interface of type IX collagen, it would be interesting to analyze if similar organization is possible in fibrillar collagens.

We have now referred to the recent paper by Boudko and Bachinger (page 8/9). Although this provides an elegant solution to the control of chain stagger in collagen IX, it is not transposable to the fibrillar collagens, for two reasons:

- (i) There is no chain stagger in the C-propeptides, unlike in the collagen IX NC2 trimer*
- (ii) There is a C-telopeptide sequence (of unknown structure) between the C-propeptide and the triple helix, unlike in collagen IX where NC2 and the triple helix are adjacent.*